

# The 2020 European Seismic Hazard Model: Overview and Results

Laurentiu Danciu[1], Domenico Giardini[1], Graeme Weatherill[2], Roberto Basili[3], Shyam Nandan[1,] Andrea Rovida[4], Céline Beauval[5], Pierre-Yves Bard[5], Marco Pagani[8,9], Celso Reyes[1], Karin Sesetyan[6], Susana Villanova[7], Fabrice Cotton[2], Stefan Wiemer[1]

[1]Swiss Seismological Service, ETH Zurich, 8009, Switzerland
[2]GFZ Potsdam, Germany
[3]INGV Rome, Italy
[4]INGV Milano, Italy
[5]Univ. Grenoble Alpes, CNRS, IRD, UGE, ISTerre, Grenoble, France
[6]Kandili, Istanbul, Turkey
[7]IST, Lisbon, Portugal
[8]GEM Foundation, Pavia, Italy
[9]Institute for Catastrophe Risk Management, NTU, Singapore

*Correspondence to*: Laurentiu Danciu (laurentiu.danciu@sed.ethz.ch)

**Abstract.**

The 2020 update of the European Seismic Hazard Model (ESHM20) is the most recent and up-to-date assessment of seismic hazard for the Euro-Mediterranean region. The new model, publicly released in May 2022, incorporates refined and cross-border harmonized earthquake catalogues, homogeneous tectonic zonation, updated active faults datasets and geological information, complex subduction sources, updated area source models, a smoothed seismicity model with an adaptive kernel optimized within each tectonic region and a novel ground motion characteristic model. ESHM20 supersedes the 2013 European Seismic Hazard Model (ESHM13, Wössner et al 2015) and provides full sets of hazard outputs such as hazard curves, maps, and uniform hazard spectra for the Euro-Mediterranean region. The model provides two informative hazard maps that will serve as a reference for the forthcoming revision of the European Seismic Design Code (CEN EC8) and provides input to the first earthquake risk model for Europe (Crowley et al., 2021). ESHM20 will continue to evolve and act as a key resource for supporting earthquake preparedness and resilience throughout Euro-Mediterranean region under the umbrella of the European Facilities for Seismic Hazard and Risk Consortium (EFEHR Consortium).



# 1 Introduction

The 2020 update of the European Seismic Hazard Model (ESHM20, Danciu et al., 2021) is the third generation of pan-European probabilistic seismic hazard models (Jimenez et al., 2003; Woessner et al., 2015). ESHM20 serves as the reference hazard model in the Euro-Mediterranean region, with the goal of supporting earthquake mitigation and resilience planning (Bisch, 2018; Santarsiero, 2018; Crowley et al., 2021). Overall, ESHM20 is developed within the seismotectonic probabilistic framework (Danciu and Giardini, 2015) incorporating both seismicity, tectonic and geological datasets and information. Within this probabilistic framework, several assumptions are made e.g., earthquake occurrence in time and space is modelled as a memory-less process requiring declustering of the earthquake catalogue to remove the dependent earthquakes, tectonic regionalization to separate regions with different geodynamical settings, seismogenic sources and active faults as proxies of future seismicity, a novel backbone ground motion model, and open access to its datasets, components and results. The exceedance probabilities of various ground motion intensity types and levels are then computed for the entire Euro-Mediterranean region using a regional hazard model that combines multiple seismogenic sources and a backbone ground motion model to account for epistemic uncertainty. The main characteristics of ESHM20 can be summarised as follows:

- It is a community-driven model, with earth science scientists, hazard and risk experts, engineers, and practitioners participating in the development phase. A core team was responsible for conducting research, coordinating the development, and collaborating closely with regional experts on a variety of tasks, from gaining access to raw data to the design and implementation of the model. Community feedback and recommendations were carefully considered via multiple channels, such as one-on-one meetings between core team members and regional experts, regional workshops, technical webinars, and conferences.

- Updated, unified and cross-border harmonized input datasets i.e., earthquake catalogues - historical (Rovida and Antonucci, 2021; Rovida et al., 2022) and instrumental (Grünthal and Wahlström, 2012; Lammers et al., 2023), tectonic info, geological faults and subduction zones (Basili et al., 2022, 2023). Extension of the model to cover the Canary Islands and Azores archipelago.

- Revisited and updated seismogenic source models including a harmonised uniform area source model and a hybrid source model that combines active faults and background smoothed seismicity. Inherent epistemic uncertainties in the earthquake rates are handled via multiple values of the parameters of a Gutenberg-Richter (GR) model, specifically $a_{GR}$, $b_{GR}$ and upper bound magnitude $M_{max}$. We also used alternative models of the magnitude frequency distributions e.g., double truncated GR distribution and a tapered Pareto distribution (Kagan and Jackson, 2000) to characterize the seismic productivity of the area sources.

- Use of the Engineering Strong Motion (ESM) database, the largest database ground motion recordings in the Euro-Mediterranean region, curated and uniformly processed as described by Lanzano et al., (2019);

- A novel ground motion model (GMM) logic tree based on the concept of a regionalized backbone approach (Kotha et al., 2020; 2022; Weatherill et al., 2020; Weatherill & Cotton, 2022; Weatherill et al., 2023). This novel approach capitalizes



upon the large ESM database and it follows the concepts initially proposed by Douglas (2018). A new ground motion predictive model for ground shaking from shallow crustal seismicity was developed and calibrated to reflect region-to-region differences in source and attenuation scaling and their epistemic uncertainties across Europe (Kotha et al., 2020, 2022; Weatherill et al., 2020). Notably, such features of the ground motion models have not been captured explicitly in earlier GMM logic trees (e.g. Delavaud et al., 2012; Danciu et al., 2018a; Pagani et al., 2020). This approach was further adapted to regions of limited data such as the stable cratonic region of northeastern Europe (Weatherill & Cotton, 2020) and the subduction and deep seismicity sources by capitalising on recent developments in regionalised ground motion modelling worldwide.

- Statistical testing and sanity checks were routinely integrated as a part of the model-building process. We used statistical tests of earthquake rate forecasts to select the declustering technique and determine the completeness magnitude-time intervals, while sanity checks were also conducted to assess the consistency of the earthquake rate forecast of individual source models and the final ensemble model. Sensitivity analyses as well as comparisons with national hazard models, were conducted as a part of the model-building development.

- The main logic tree combines both the earthquake rate forecast and backbone ground motion model and it was optimized for large-scale computation of the ground shaking hazard depicted by Peak Ground Acceleration (PGA) and a pseudo-acceleration spectrum (SA) with 5% damping at fifteen spectral ordinates from 0.05s to 5s. The horizontal component is described by the $50^{th}$ percentile of the response spectra of tow horizontal components projected onto all non-redundant azimuths i.e., *RotD50* (Boore et al., 2010, Kotha et al., 2020)

- Complex computational models and pathways were implemented in OpenQuake hazard engine (Pagani et al 2014), which was continuously enhanced throughout the model development cycle. The computational input files are released publicly under an open Creative Commons license (see https://creativecommons.org/, last accessed December 2023), which allows users to reproduce the results and adapt the model for different needs and applications.

Furthermore, during the model development we established a close collaboration with a working group from the Technical Committee of the European Committee for Standardization (TC250/ CEN). This working group is responsible for updating the seismic action specifications within the Eurocode standards (SC8). During its final development cycle between 2020 and 2021, experts from Albania, Bulgaria, Croatia, France, Finland, Greece, Germany, Italy, Iceland, Portugal, Romania, Slovenia, Sweden, Switzerland, Turkey, Norway evaluated and tested the ESHM20 components and outcomes. Although no final agreement had been reached at the time this manuscript was submitted for publication, the convener suggested to SC8 that the two ground-shaking hazard maps derived from ESHM20 will be included in the next generation EN1998-1-1 Annex G (Labbé and Paolucci, 2022). These maps will serve as *informative reference* of ground shaking values across the Euro-Mediterranean region as part of the next update of the European seismic design code (CEN EN1998-1-1) in the region.

ESHM20 underwent a release cycle that included a community preview version, a beta version, and a final version that was made available in December 2021 and publicly released in May 2022. The collaboration between the model core team,



community members, and communication experts was essential to this release, with the goal of making outreach products to efficiently communicate earthquake-related hazards and risk models and results. The approach used to make the hazard and risk model accessible and easy understandable by the general public drew upon communication concepts derived from mass media, including communication principles, key messages, specific products and target audiences. Two surveys were conducted to ensure that these outreach products met the needs of end-users, with a focus on the interactive hazard and risk web-viewer and specialized posters illustrating the European earthquake hazard and risk maps (Dallo et al., 2023). The outreach materials are available in various languages.

In this contribution, we provide an overview of the background, scientific and collaborative framework, computational aspects, and key results of ESHM20. Many aspects of the updated hazard model are presented and discussed in the model's Technical Report (Danciu et al., 2021) and the lessons learned in Danciu et al. (2022), while the remaining components are part of this special issue. Companion manuscripts describe the historical catalogue (Rovida et al 2022), the ground motion models (Weatherill et at 2023) and overall model implementation and documentation (EFEHR Technical Report, Danciu et al 2021). A description of the active fault database and subduction systems is described by Basili et al. (2022). We begin by illustrating and discussing the key differences between ESHM20 and its precursor ESHM13 (Giardini et al., 2014) in Section 2. In Section 3, we describe key input datasets underpinning ESHM20. Subsequently, in Section 4 we describe details of the spatial-temporal earthquake rate forecasts and the methodologies adopted. Section 5 provides a concise overview of ground motion models, while Section 6 explores the primary logic tree and computational aspects in more detail. The latter section delves into the main findings of ESHM20, including the contributions to hazard from different seismogenic source models, as well as the associated range of uncertainties. Finally, Section 7 summarises the main outcomes and offers an outlook for future research directions. Overall, this manuscript aims to describe the key features of the model, its results, and their accessibility. Readers are encouraged to access the ESHM20's input datasets, documentation, and computational files on the publicly accessible repository listed under "Datasets and Online Resources" Section in this manuscript.

## 2 ESHM20 (Danciu et al 2021) vs ESHM13 (Wössner et al 2015): What is different?

Regional seismic hazard models are evolving models that progress over time, with the most recent models based on updated input datasets and the most recent scientific outcomes. This is obvious when comparing, for instance the ground motion datasets underlying some of the ground motion models used in ESHM13 (Delavaud et al., 2012; Woessner et al., 2015), with an average of about 500 strong motion records used, and more recent GMMs. Nowadays, the significantly larger datasets, make empirical ground motion models more robust (Kotha et al., 2020, 2022; Lanzano et al., 2019, 2020; Manea et al., 2022; Kuehn et al., 2019) and support the development of regional adjustments and modelling of epistemic uncertainties (Weatherill and Cotton, 2020; Douglas, 2018; Kowsari et al., 2023; Lavrentiadis et al., 2023). Similarly, the updates on the earthquake catalogues and revisiting the seismogenic source models must be accounted for as a potential source of differences between



the two models. The spatial distribution of the PGA for a return period of 475 years of both ESHM13 and ESHM20, have similar spatial patterns and the difference map given in Figure 1, illustrates how the $PGA_{ESHM20}$ and $PGA_{ESHM13}$ mean values

differ at each location. Overall, when compared with the ESHM13 the PGA values of ESHM20 for a return period of 475 years have been decreased in various regions. There are certain regions in Albania, Greece, southern Portugal, southern Spain, and western Turkey that exhibit lower values when compared with the ESHM13. Such differences are the result of various factors including updated input datasets, updated seismogenic source models, updated subduction sources, updated slip-rates and maximum magnitude of the active fault and a different source model logic tree (Danciu et al., 2021, 2022).

**Figure 1: The top panel highlights regions where the mean peak ground acceleration PGA values for ESHM20 increased (shown in**
**red) or decreased (shown in blue) as compared to ESHM13. The bottom panels show the spatial distribution of the mean PGA for a return period of 4745 years for ESHM13 (lower left) and ESHM20 (lower right). The maps are both estimated for a generic rock site class with a shear wave velocity $V_{s30}$~800m/s.**



Indeed, all these changes resulted in an updated earthquake rate forecast. Comparison earthquake rate forecast maps between the two versions of the European hazard models are given in the *Supplementary Materials.* Moreover, the changes to the ground motion models are significant due to the way in which epistemic uncertainties were handled, with a paradigm shift from multiple ground motion models (Delavaud et al., 2011) to the regional backbone approach (Douglas, 2018; Weatherill et al., 2020). Weatherill et al. (2020) compare the ground motion of the two regional hazard models for the active shallow crust and find that the ESHM20 models compute lower values for very short spectral periods (PGA and PSA 0.1s) than those used in ESHM13; differences also occur at greater distances, where the general trend for ESHM20 is towards faster attenuation. Consequently, the core and body of the ESHM20 regional backbone logic tree predict smaller motions than the ESHM13 ground-motion logic tree. It is worth noting that the ground motion models of ESHM13 are based on datasets of ground motion recordings from outside of Europe. These recordings may have a different attenuation rate, reflect unique source characteristics, or exhibit site conditions not found in the ESM Database (Lanzano et al., 2019). In contrast, the increased number of strong motion records from the Apennine region in Italy shifts the centre of the strong motion dataset toward conditions that are predominantly more rapidly attenuating than the rest of Europe. Comparison plots between the ground motion models used in ESHM20 and ESHM13, respectively are also given in the *Supplementary Materials*. Additional factors that may also contribute to these differences in weighting schemes applied to the two models, as well as the model implementation in OpenQuake (Pagani et al 2014). In ESHM13, the weighting scheme is return-period dependent, and applied in post-processing, whereas in ESHM20 the weights are applied to each branch. A sampling technique of the entire logic tree was then used to obtain the results of ESHM20. To conclude, the transition from ESHM13 to ESHM20 represents the continuous effort in regional seismic hazard advancement, and ESHM20, with its updated datasets, methodological enchantments, and comprehensive model integration, now stands as the reference model for the region, superseding ESHM13.

## 3 Main Input Datasets

ESHM20 is based on the integration of multidisciplinary datasets and expert information. The main datasets include the unified earthquake catalogue, both historical (Rovida and Antonucci, 2021; Rovida et al., 2022) and instrumental (Lammers et al., 2023) earthquake catalogues, shallow active faults and subduction systems (Basili et al., 2022), and ground motion recordings (Lanzano et al., 2019). All data sets were meticulously collected, uniformly processed, and harmonized across pan-Euro-Mediterranean. However, due to the heterogeneous seismogenic characteristics across the Euro-Mediterranean region, the compiled datasets exhibited variations in completeness, both spatially and temporally. This inherent variability in the quality of data presented significant modelling challenges for both seismogenic source and ground motion characterization, ultimately influencing the ESHM20's overall uncertainties and outcomes. A tectonic regionalization of the Euro-Mediterranean was developed using the information and data from the ESHM13 and NEAMTHM18 models (Basili et al., 2021). The regionalization consists of eleven tectonic domains, namely active volcanoes, back-arc and orogenic collapse, continental rift,



oceanic rift, contractional wedge, accretionary wedge, conservative boundary, transform zones proper, shield, stable
continental region, and stable oceanic region. This regionalization is the basis for the TECTO layer, which is later used for
organizing the spatial consistency of the seismogenic sources.

**European Fault-Source Model 2020** (EFSM20, Basili et al., 2022) is a compilation of the existing regional active fault
databases. EFSM20 includes 1,248 crustal faults spanning a total length of 95,100 km and four subduction systems, namely
the Gibraltar, Calabrian, Hellenic, and Cyprus Arcs. The fault model covers a region with a buffer of 300 km around all
180 European countries (except for Overseas Countries and Territories) and a maximum depth of 300 km for the subducting slabs.
All parameters (i.e., fault trace, geometry, depth, length, width, strike, dip, rake and slip rate values) needed to develop a
seismogenic source model were estimated for crustal faults and subduction systems (Basili et al., 2023).

**Unified Earthquake Catalogue** consists of two parts: the European PreInstrumental Earthquake Catalogue EPICA v. 1.1
(Rovida and Antonucci, 2021; Rovida et al., 2022), with seismicity occurred between the years 1000 CE and 1899 and the so-
185 called instrumental earthquake catalogue (with seismicity the took place after 1900) based on the updated EMEC catalogue
(Lammers et al., 2023). **EPICA v1.1** included earthquake data from 1000 to 1899 CE and builds upon the latest knowledge
gathered in the European Archive of Historical Earthquake Data (AHEAD, Albini et al., 2013; Locati et al., 2014; Rovida and
Locati, 2015). It contains 5703 earthquakes with maximum observed intensity above 5.0 or Mw larger than 4.0. It is based on
160 macroseismic datapoints (MDPs) sources and 39 regional catalogues selected from AHEAD. These datasets were
considerably updated with respect to those used for compiling the historical catalogue for ESHM2013 (Stucchi et al., 2013).
A systematic analysis of these data resulted in the selection of the most representative description of each earthquake,
independently from national constraints. For 3297 earthquakes, parameters are newly assessed from MDPs using harmonized
procedures based on the attenuation of macroseismic intensity from macroseismic datapoints (MDPs), with the goal of ensuring
consistency of earthquake parameters across countries. These parameters are then combined and integrated with parameters
harmonised from recent regional catalogues. The instrumental earthquake catalogue spans from 1900 to 2014 CE and updates
the earlier European-Mediterranean Earthquake Catalogue (EMEC), constructed and adapted for use in the ESHM13 by
Grünthal et al (2013). EMEC compiles earthquake data from various local, national and international earthquake bulletins,
along with special studies for particular regions or earthquake sequences. The data are then harmonised by applying region-
specific hierarchies that identify the preferred earthquake source from those compiled for each event. Using region-specific
and magnitude-scale conversion relations, earthquake magnitudes are rendered into a common reference magnitude scale,
equivalent to moment magnitude ($M_w$). During the compilation process, the European catalogue was aligned with existing
harmonised national catalogues where possible, which along with special studies were prioritised over local and national
bulletins. The updated EMEC extends the most recent period of coverage that previously ended in 2006 (Grünthal & Wahlstöm,
2012; Grünthal et al., 2013) and incorporates recent national catalogues such as the Catalogo Parametrico di Terremoti Italiani
(CPTI) the 2017 French Seismic Catalogue (F-CAT, Manchuel et al., 2018), among others. The instrumental earthquake
catalogue now contains over 55,700 shallow and deep earthquakes with Mw greater than 3.5 (Lammers et al., 2023). We





further analyzed the unified earthquake catalogue to identify and remove the foreshocks and aftershocks as a prerequisite of Poisson assumption for the earthquake recurrence rates. We performed comparisons amongst various declustering techniques, including the time-space windows method (Gardner and Knopoff, 1974; Uhrhammer, 1986; Grünthal, 1985), cluster method

(Reasenberg, 1985) and the spatial-time correlation metric-based declustering (Zaliapin et al., 2008). The cluster method retained the highest number of events post-declustering, surpassing both Zaliapin's approach and the standard time-space window technique (Grünthal, 1985) used in ESHM13. Additional spatial analyses i.e., comparison of events per 50 by 50km grid cells, as well as Information Gain tests, suggested that the cluster method (Reasenberg, 1985) and an alternative to the declustering method (Grünthal, 1985) used in ESHM20. Initially, two alternative declustering methods were part of the logic

tree of the seismogenic sources of the pre-release model. However, due to overall computational increase, we reduced complexity of the model, and this branch was removed after conducting several sensitivity analyses. In addition, there were open discussions within the scientific community about the implications and challenges associated with declustering. Numerous studies (Beauval et al., 2006, 2013; Sesetyan et al., 2018; Marzocchi and Taroni, 2014; Marzocchi et al., 2020; Meletti et al., 2021; Mizrahi et al., 2021; Taroni and Akinci, 2021) have explored the effects of declustering and implications

to seismic hazard modelling. These studies suggest that the use of declustering could lead to reduced seismicity rates, potentially resulting in underestimated hazard estimates, especially in regions characterized by seismic swarms and prolonged aftershock sequences. However, we acknowledge the ongoing need for further research to explore this topic in greater depth, as well as time to reach a community consensus; nonetheless our approach closely adheres to current practices in the field (Gerstenberger et al., 2020). Likewise, the assessment of the catalogue completeness is arguably as crucial, if not more so,

than the choice of the declustering algorithm. The key parameter in this case is the magnitude of completeness (Mc), defined as the lowest value of magnitude at which all events are detected in space and time. Mc is variable in space and time, and – in general terms - it decreases with time given the increase in the density of seismic stations. To effectively accomplish this in ESHM20, a new algorithm that combines the Temporal Course of Earthquake Frequency (TCEF) approach (Nasir et al., 2013), with the Maximum Curvature method (Woessner and Wiemer, 2005; Wiemer and Wyss, 2000) was proposed. This innovative

method involved an iterative process beginning with the declustering of the earthquake catalogue, followed by dividing it into various magnitude intervals, and generating cumulative earthquake time series for each interval. The time-series were used to visualize and detect the change points in the catalogue's completeness. Next, statistical tests were used to validate the findings. Finally, the results of this procedure were the Mc values and completeness time-intervals for the completeness super zones (CSZs) of ESHM20. Details are given in Section 2.2 of the ESHM20 Technical Report Danciu et al. (2021).

**4 Seismogenic Source Models: Spatial-Temporal Earthquake Rates Forecast**

The ESHM20 seismogenic source characterisation builds upon the ESHM13 legacy and it comprises four distinct source models to capture the spatial and temporal variability of earthquake occurrence across the pan-European region:





- Unified Area Sources Model: This model updates the ESHM13 area source model by incorporating new area sources of the recent national seismic hazard models and considering the feedback and opinions of the local experts. More specifically, the model incorporates contributions from the recent earthquake hazard models from Italy (Meletti et al., 2021; Visini et al., 2022) , Germany (Grünthal et al., 2018), Spain (Rivas-Medina et al., 2018; Benito Oterino et al., 2012), the United Kingdom (Mosca et al., 2022), Slovenia (Šket Motnikar et al., 2022), Switzerland (Wiemer et al., 2016), Turkey (Sesetyan et al., 2018), Iceland (Halldórsson et al., 2022), France (Baize et al., 2013), Bulgaria (Solakov et al., 2014), Finland (Fülöp et al., 2023), Balkan region (Mihaljević et al., 2017) and Romania (Vacareanu et al., 2016). Further, cross-border harmonization (i.e., removing the overlapping area sources at national borders) is applied to all sources, following Danciu et al., 2018b; Pagani et al., 2020. This source model includes regions of homogenous seismicity classified as shallow crustal, volcanic, subduction in-slab, and deep seismicity (e.g., Vrancea region, Romania). They define regions with a known history or potential for seismic activity but not necessarily all associated with specific faults (see Figure 2a).

- Hybrid Model of Active Faults and Background Smoothed Seismicity: The model integrates active faults with background seismicity in regions where the active faults are identified; otherwise, a smoothed seismicity, grid-based is used. An adaptive kernel was regionally calibrated and used for smoothing the seismicity. In Figure 2b, the active faults (red lines) are primarily observed along the tectonic boundary that extends through the Mediterranean region. This would likely include countries like Greece, Turkey, Italy, and regions along the coastlines of the Adriatic, Aegean, and Ionian Seas.

- Subduction Sources: These sources represent the subduction interface and in-slab seismicity of the Hellenic, Cyprian, Calabrian, and Gibraltar Arcs.

- Non-subducting Deep Seismicity Sources: This model accounts for nested seismicity with depth in Vrancea, Romania and the cluster of deep seismicity in the southern Iberian Peninsula.

Earthquake activity rates for individual seismogenic sources are estimated under the assumption that regional seismicity follows a memoryless Poisson process. This process is characterized by a stationary mean rate of occurrence defined by a Gutenberg-Richter (GR) model and its main parameters (aGR and bGR). The activity rate parameters are estimated for large-scale tectonic domains (TECTO zonation layer) and the zonation-based area-sources model (ASM) using the declustered unified earthquake catalogue and the magnitude-time completeness intervals. The TECTO zonation layer as shown in Figure 2b, is used as a proxy of the spatial organization of the source models and provides a basis for estimating more stable GR parameters given the larger number of earthquakes within each super zone. The seismic productivity of the active shallow faults is calculated by converting long-term geological or geodetic slip rates into activity rates under the assumption of moment conservation (Anderson and Luco, 1983; Bungum, 2007; Danciu et al., 2018b).



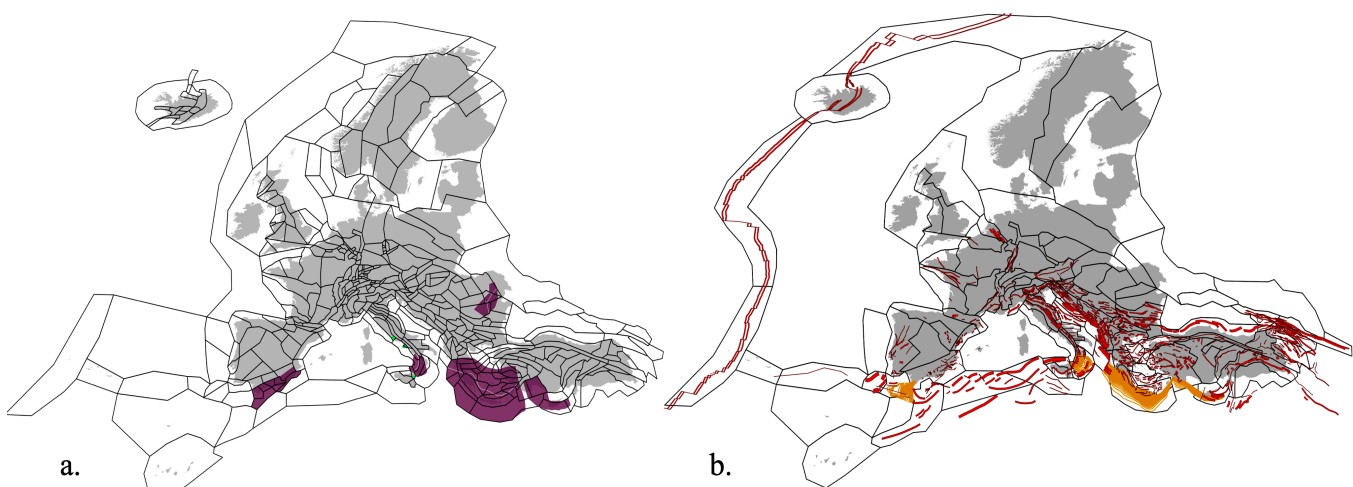

a.                                b.

**Figure 2: Schematic illustration of the ESHM20' seismogenic source models: a) shallow crust area sources (black), volcanic area sources (green), deep seismicity and subduction in-slab (purple); b) shallow crust active faults (red lines), subduction interface (orange polylines). Tectonic zonation is highlighted in black polygons.**

For each fault, three slip-rates (converted to aGR), three $M_{max}$ values, a single regional bGR-value (from TECTO) and the average fault area are considered. The $M_{max}$ values of each fault source are obtained from empirical magnitude – fault geometry

scaling relationships (Leonard, 2010, 2014) and take into account the full range of uncertainties of the fault geometries and the scaling relations. Similarly, the empirical equations of Allen and Hayes, 2017 were used to estimate the $M_{max}$ for subduction interface sources; the activity parameters were obtained from a rather complex logic tree with more than 2800 end-branches. For area sources, the $M_{max}$ is updated following the strategy of ESHM13. The lower value of the expected $M_{max}$ is calculated using the highest magnitude value observed for that tectonic regime, including the average standard error of that magnitude in

the entire earthquake catalogue. The second and third values of $M_{max}$ add 0.3 and 0.6 magnitude increments to the lower bound magnitude value, respectively. Finally, two types of magnitude-frequency distributions (MFDs) are used to characterize the earthquake recurrence models for the area sources: a double truncated GR and a tapered Pareto distribution (TPD, Kagan and Jackson, 2000).

Statistical testing and sanity checks are critical components of the model-building process, particularly when reaching scientific

consensus is difficult (Meletti et al., 2021). We performed statistical analyses to assess earthquake rate forecasts within the seismogenic source model at each development stage. Sanity checks were routinely conducted by comparing the forecasted total number of earthquakes across different magnitude bins ($M_w \sim$ 4.5 to 8.5) against observed number of events within various tectonic domains. These sanity checks helped us to identify and address discrepancies in the forecasts and reconcile them when possible. Additionally, retrospective testing was conducted by estimating the information gain, which quantifies the differences

in performance between earthquake rate models using only the observed $M_w \geq 6$ earthquakes from the unified earthquake



catalogue. Again, the information was used to assess the performance of each individual source model, as well as of the ensemble earthquake rate forecast.

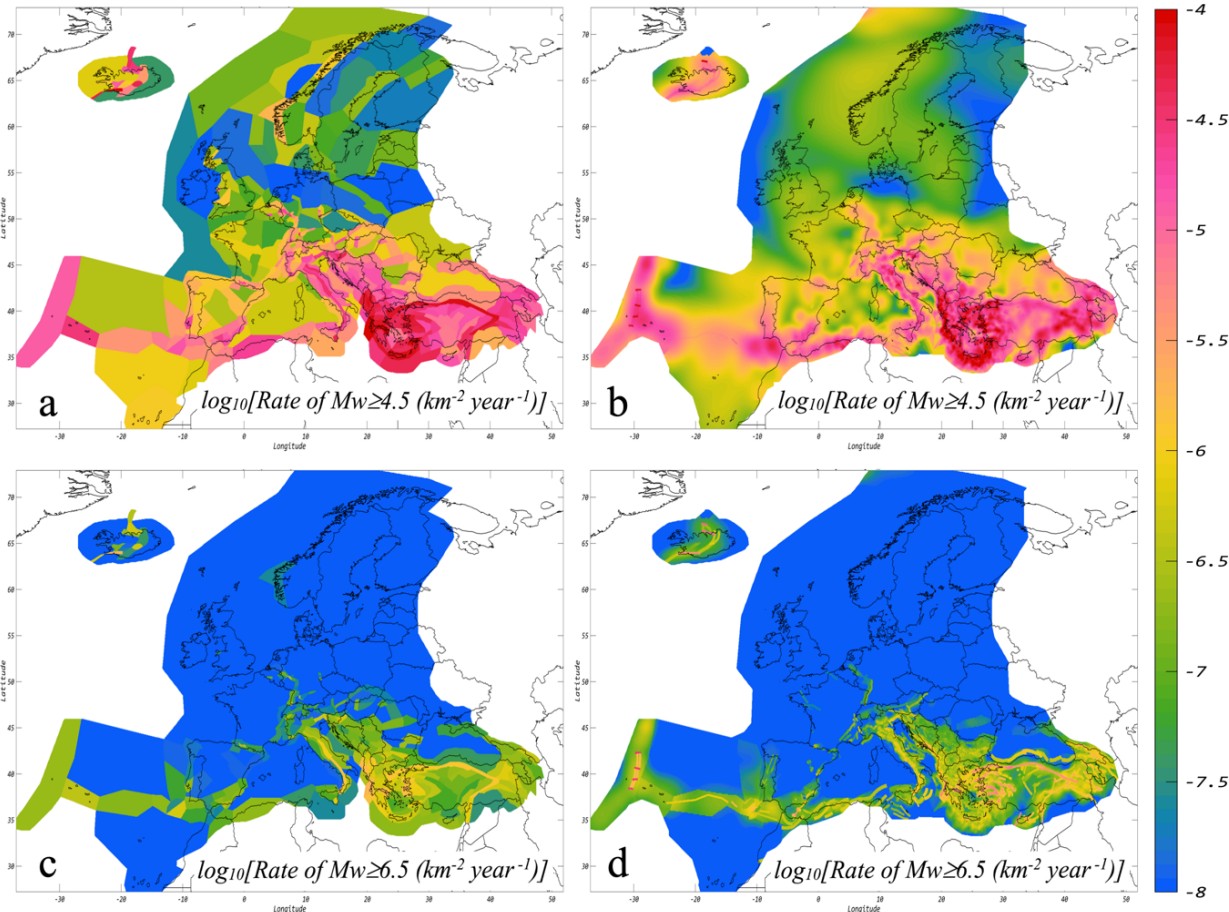

**Figure 3: Spatial distribution of annual recurrence rates in log units as forecasted by the ESHM20' seismogenic source models.**
**Panel (a) displays the annual rates for events with M>4.5 forecasted by the shallow area sources, while panel (b) shows those from the hybrid model incorporating active shallow crust faults and background smoothed seismicity. Similarly, panel (c) presents the annual rates for events with M>6.5 from shallow area sources, and panel (d) from the hybrid model with active shallow faults and background smoothed seismicity. All rates are normalized per unit of area.**

Figure 3 illustrates the spatial distribution of the forecasted annual rates of the two main source models. Panel (a) displays the rates for events with $M_w \geq 4.5$ from shallow area sources, while panel (b) shows those from the hybrid model incorporating active shallow faults and background smoothed seismicity. Similarly, panel (c) presents the rates for events with $M_w \geq 6.5$ from shallow area sources and panel (d) from the hybrid model with active shallow faults and background smoothed seismicity. It is worth noticing, that the earthquake rate forecasts across the two primary source models of the ESHM20 model exhibit

remarkably consistent trends, though the spatial variability of the earthquake rates is still evident. For instance, in the case of



$M_w\geq4.5$, the spatial variability of the earthquake rates is more clearly driven by the shapes of specific source areas, as opposed to the more diffused pattern observed in the smoothed seismicity. Similarly, for magnitudes greater than $M_w\geq6.5$, the consistency of the spatial pattern of the forecasted earthquake rates is even more evident. This is a direct consequence of a deliberate modelling decision to use active faults as spatial proxies in the delineation of area sources, as seen in panels c and 310 d of Figure 3. However, the forecasted earthquake rates tend to be higher for active faults in comparison to those of area sources. ASM forecasts a net yearly rate of $M_w\geq4.5$ of about 91.60 and 1.26 for events with $M_w\geq6.5$. On the other hand, hybrid rate forecasts (active faults and smoothed seismicity) forecast a net yearly rate of $M_w\geq4.5$ of about 82.40 and 2.03 for $M_w\geq6.5$, respectively.

**5 Ground Motion Characteristic Models: Synopses**

The ground motion model (GMM) used in the ESHM20 incorporates regional features of the seismic source, path and site conditions, as well as aleatory (random) and epistemic (model-to-model) uncertainties. Epistemic uncertainty is typically captured using multiple ground motion models selected from the literature and applied as alternative branches in a logic tree (Delavaud et al., 2012; Danciu et al., 2018; Lanzano et al., 2020). However, the widely used multiple model approach has some limitations, such as the tendency for selected models to share common underlying data leading to similar predictions for 320 well-represented scenarios, inconsistent parameterizations between models and cases where epistemic uncertainty may be under-estimated owing to a lack of suitable models in the literature. To address these issues, in ESHM20 we adopt a scaled backbone approach that develops on ideas first suggested by Douglas (2018). For shallow crustal seismicity a suitable backbone GMM is developed from the large dataset of ground motion observations in the ESM database (Lanzano et al., 2019). With this data, region-specific adjustments are constrained within the regression and their corresponding distributions applied 325 in the hazard calculation to capture epistemic uncertainty. These adjustments account for region-to-region variability in the model and allow for a more complex representation of the underlying uncertainty. The use of region-specific adjustments also enables the transition from an ergodic to a non-ergodic model, where systematic effects in specific regions are identified and extracted from the aleatory variability. This process helps reduce aleatory uncertainty and allows new data to refine the model calibration over time. The ESHM20 backbone model for active shallow crust is built upon the regional ground motion model 330 of Kotha et al.(2020; 2022), with the development of the ESHM20 GMM logic tree proposed by Weatherill et al., (2020) and the final details of the implementation given in Weatherill et al., 2023.

The purely data-driven approach for the development and calibration of the *regionalised, scaled backbone-logic tree* cannot be applied as-is to all regions of Europe as data is limited or absent in some cases. For example, in the tectonically stable *Cratonic* region of north-eastern Europe the seismological properties of the crust differ significantly from those of southern 335 and western Europe, warranting a different approach to ground motion characterisation and its epistemic uncertainty. Weatherill and Cotton, 2020 propose a GMM logic tree for application in north-eastern Europe. They do so by first analysing



different geophysical datasets for Europe to demonstrate the close analogy between the stable Shield region of the Baltic Sea and its surrounding countries and the tectonically stable crust of the Central and Eastern United States (CEUS). Then, leveraging upon the suite of GMMs recently developed for the CEUS as part of the Next Generation Attenuation East (NGA

East) Project (Goulet et al., 2021), they fit a modified form of the Kotha et al. (2020) to the distribution of expected ground motions predicted by the different NGA East GMMs. The model-to-model variability, $\sigma_\mu$, is used to quantify the scaling factors to be applied to this new backbone to represent the epistemic uncertainty in the expected ground motions for this low seismicity region.

The regionalized backbone GMM concept was further expanded to encompass deeper seismic activity, specifically in regions

such as Vrancea, Romania, as well as subduction systems including the Hellenic, Cypriot, Calabrian and Gibraltar Arc. Given the limited availability of ground motion data pertaining to earthquakes occurring in subduction zones, the development of a new GMM for subduction and deep seismicity was not straightforward. The so called "BC Hydro" model of Abrahamson et al. (2016) has been identified as a suitable backbone GMM for the subduction regions. This GMM fulfils the necessary criteria of the project and exhibits consistently good fit to observed data in the ESM database across various spectral periods in each

of the three subduction regions. Moreover, it was determined that the Abrahamson et al. (2016) ground motion model (GMM) for subduction in-slab events is also applicable to the Vrancea deep non-subduction zone.  Using the ground motion data for these regions included in the ESM database for small-to-moderate magnitude earthquakes, we adjust the anelastic attenuation term to align it with the seismological properties of the eastern Mediterranean; however, the narrow magnitude range of recordings prevent us to recalibrate the source scaling selected GMM.  Additionally, the BC-Hydro GMM includes a

forearc/backarc scaling term with faster attenuation for backarc sites and a trend also presents in the ground motion data from the Hellenic Arc and Vrancea deep seismic zones. While the forearc/backarc scaling coefficients were not themselves revised owing to the limited data set in the backarc regions, further modifications were made to the BC Hydro model to allow a smoother transition from the forearc to the backarc regime, thus preventing sudden drops in hazard across the forarc/backarc margin. Scaling factors to capture epistemic uncertainty in source stress parameter and anelastic attenuation coefficients for

the subduction and deep seismicity GMM were calibrated from analysis of region-to-region variability = across different subduction zones worldwide, following observations from the NGA Subduction project. The complete description of the development of the scaled backbone GMM logic tree for subduction and deep seismicity is found in Weatherill et al. (2023).

## 6 ESHM20: Main Logic Tree and Computational Aspects

The epistemic uncertainties in ESHM20 are handled in a logic tree framework, reflecting the current state-of-practice in seismic

hazard assessment. Given the size of the computational region and the complexity of the two sub-components of the logic tree, we aimed at a balanced logic tree structure where both the epistemic uncertainty of the individual source models and the ground motions models are adequately represented. In ESHM13, due to combined factors such as the complexity of the input models,



large scale geographical grid, size and volume of the hazard results, and software and hardware constrains, a collapsed earthquake rate model was used. This was combined with the multiple GMPEs in a postprocessing step to estimate the weighted
mean and the weighted quantiles of ESHM13. A consequence of this decision is that the quantiles range of the results is very narrow in many parts of the model. Hence, in ESHM20, we aimed to capture the body, centre and range of the expected ground motion. The main logic tree combines the two main components i.e., the seismogenic sources and the ground motion models as illustrated in Figure 4. The former, given in Figure 4a, consists of four branching levels, corresponding to a source type branching level and three more branching levels depicting the epistemic uncertainty of the type of magnitude frequency
distribution, activity parameters (Gutenberg-Richter parameters i.e., aGRs, bGRs) and upper magnitudes ($M_{max}$). A key aspect was related to the use of correlated or non-correlated uncertainties of the activity rate parameters. The implementation of uncorrelated uncertainties is suitable for site-specific hazard analyses, while they pose computational challenges at the regional level. For instance, having a source model with three fully correlated branches for aGR, three for bGRand three for $M_{max}$ would result in 9 logic tree branches in total. However, the upper and lower branches would represent extreme cases, leading to
exceptionally high or low activity rates. On the other hand, uncorrelated uncertainties would require permuting all combinations of aGR, bGRand $M_{max}$ for all sources, resulting in an exponentially large number of realizations (i.e., 9 end-branches and 400 area sources will result in $9^{400}$ uncorrelated end-branches) that makes it impracticable for large-scale regional models like Euro-Mediterranean region. Sensitivity analyses were conducted, and for the model implementation we considered the correlated end-branches for the seismogenic source model. The epistemic uncertainty of the ground motion is extensively
described by Weatherill and Cotton, 2020 and Weatherill et al., 2020 and its development involved several iterations. Initial tests of the complete seismic hazard model were followed by minor modifications to the GMM logic trees (see Figure 4b) for shallow seismicity in ESHM20. The original logic tree structure was preserved, with adjustments made to improve the representation of epistemic uncertainty. Specifically, the number of branches for source region uncertainty was increased to five, while the three-branch representation for attenuation region uncertainty was retained. This approach was also applied to
the logic tree case for the cratonic sources. A higher numbers of branches were considered but were found to be computationally prohibitive (Weatherill et al., 2023a). Practical decisions were made to streamline computational time and make the best use of hardware resources. These decisions include using a single branch for smoothed seismicity and the subduction interface and a reduced number of grid points, particularly offshore. Other optimizations included using point ruptures, weighted depths for sources and the dominant style of faulting. For a comprehensive representation of epistemic
uncertainty, we used the random sampling technique in OpenQuake to sample 10,000 logic tree end-branches. Noteworthy, the OpenQuake features open-source libraries and supports standardized, backward-compatible file formats. The configuration file for the ESHM20 calculations in OpenQuake, along with the main input files, are publicly available (see Data Resources Section) and the main results are presented in the next section.






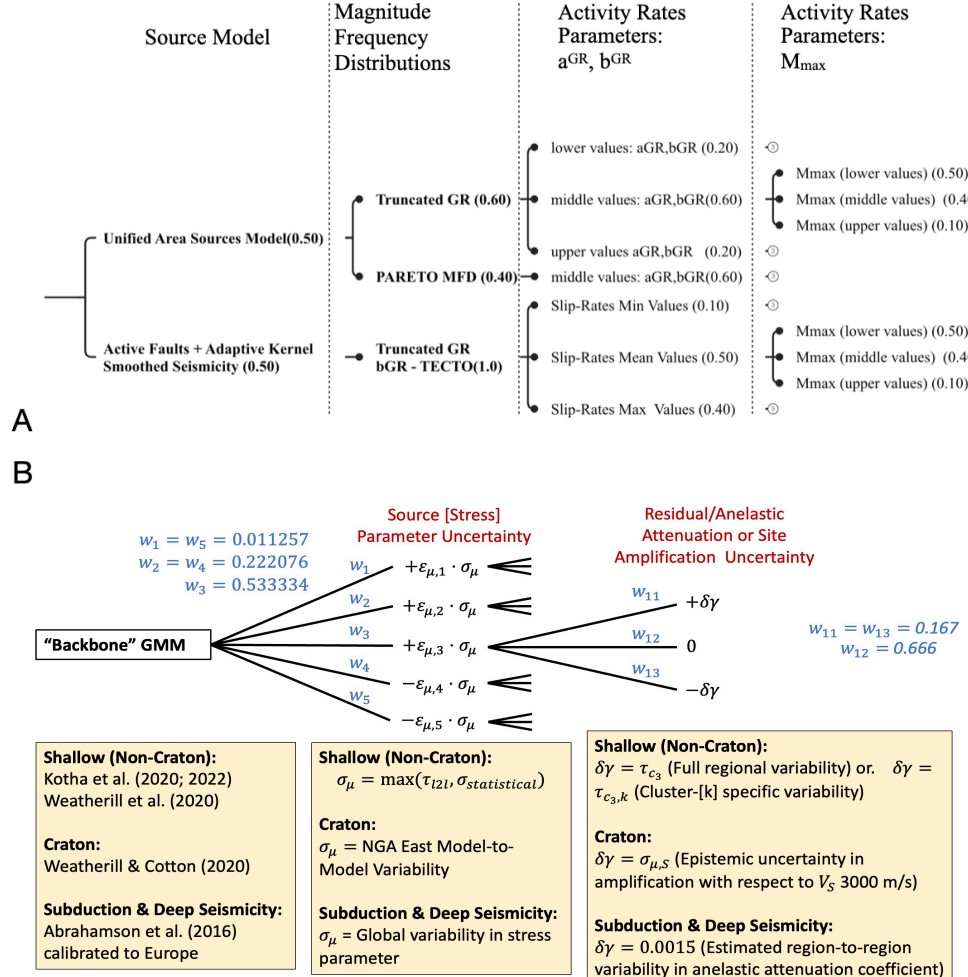

**Figure 4: Schematic overview of the main logic tree for shallow crust regions as defined for the ESHM20 computational model. Panel A illustrates the asymmetric branching structure of the seismogenic source models. Panel B presents the branching levels of the Ground Motion Models, highlighting both the default and cluster-specific scaled backbone models as implemented by Weatherill et al., 2023.**

## 6.1 ESHM20: Main Results

The ESHM20 results provide time-independent earthquake ground-shaking exceedance probabilities described as hazard curves, uniform hazard spectra and hazard maps. Mean, median (50[th] percentile) and four key quantiles (5th, 16th, 84thand 95th) are provided for various intensity measure types: PGA and spectral acceleration (SA) over a range of fundamental periods (0.05 to 5 seconds) with 5% damping. The hazard maps and the UHS are reported for five annual probabilities of exceedance (APEs) i.e., 0.02, 0.0021, 0.001, 0.000404 and 0.0002 which correspond to five return periods i.e., 50, 475, 975, 2475 and 5000 years, respectively. The ESHM20 results are valid for *RotD50* of the horizontal components (Boore et al., 2006; Boore, 2010) as provided by the selected ground motion models and estimated for a uniform rock site condition of $V_{s30}$ ~ 800 m/s.



Hazard curves are calculated up to extremely low annual probabilities (e.g., $10^{-4}$, which corresponds to return periods of 10000 years return periods). However, a degree of caution is required when interpreting curves at these low probability levels, where the aleatory uncertainties of the ground motion, as well as the robustness of the earthquake rates must be carefully analysed, similarly with a site-specific hazard modelling. Therefore, hazard maps are limited to a maximum APE of 1/5000 or an equivalent return period of 5,000 years to address these challenges. Figure 5 illustrates the hazard curves, mean and five

quantiles (5th, 16th, 50th, 84th, and 95th) for Bucharest (Romania), Istanbul (Türkiye), Zagreb (Croatia), Syracuse (Italy), Lisbon (Portugal) and Stockholm (Sweden).

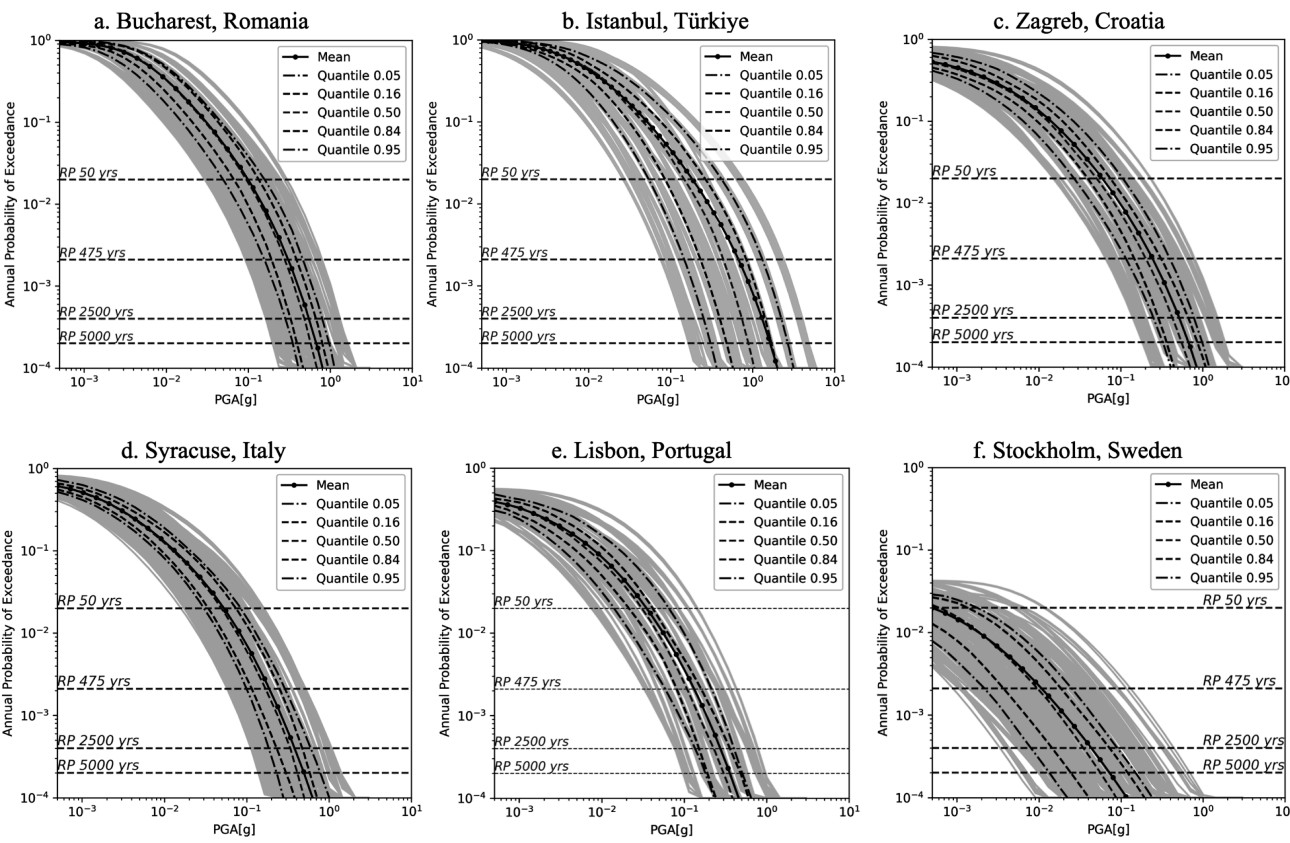

**Figure 5: Full distribution of hazard curves for 10000 samples of the ESHM20 main logic tree, mean and five quantiles (5th, 16th, 50th, 84th, 95th are given for Bucharest (Romania), Istanbul (Türkiye), Zagreb (Croatia), Syracuse (Italy), Lisbon (Portugal) and**

**Stockholm (Sweden). These results are applicable to generic rock soil conditions with a shear wave velocity (Vs30) of 800m/s.**

The quantiles were obtained from a distribution of hazard curves (in grey lines in Figure 6) corresponding to 10000 end-branches samples of the main logic tree given in Figure 4. The body and range of hazard curves depict the combined uncertainties of both the seismogenic sources and ground motion at each location. Bucharest and Syracuse exhibit a similar range of hazard estimates and a narrow quantile range. Notably, at these locations, the configuration of the logic tree is

different, combining more than a single seismotectonic environment i.e., active shallow and non-subduction deep seismicity



of Vrancea for Bucharest; for Syracuse, a complex combination of active shallow, subduction interface, in-slab of the Calabrian Arc, as well as volcanic seismicity. Similarly, Istanbul is subjective to shallow active seismicity, in particular to the very seismically active faults of the North Anatolian Faults System. Zagreb is primarily exposed to the shallow seismicity, while Lisbon is affected by shallow seismicity both on and offshore, including the southern offshore source responsible for the 1755

earthquake. The range of seismic hazard curves depicting the end-branches realizations is wide for Stockholm when compared with the other locations. This wide distribution depicts the complexity of the ground motions characteristic models considered for this Craton region i.e., the scalable regional backbone logic tree (Weatherill and Cotton, 2020; Goulet et al., 2021),as well as the contribution from the area source and the smoothed seismicity branches. Furthermore, the hazard curves at different location have different decay within the range of APE of interest from 1/50 to 1/5000. The decay of the hazard curves, indicates

changes in hazard estimates: a fast decay results in high hazard values, whereas a slow decay indicates a low hazard value at given APE. In comparison with other locations, Istanbul and Bucharest exhibit high seismic hazard values across all APEs as indicated by the faster decay of the hazard curves, while the decay of the hazard curves for Stockholm is the slowest, resulting in the lowest hazard PGA values.

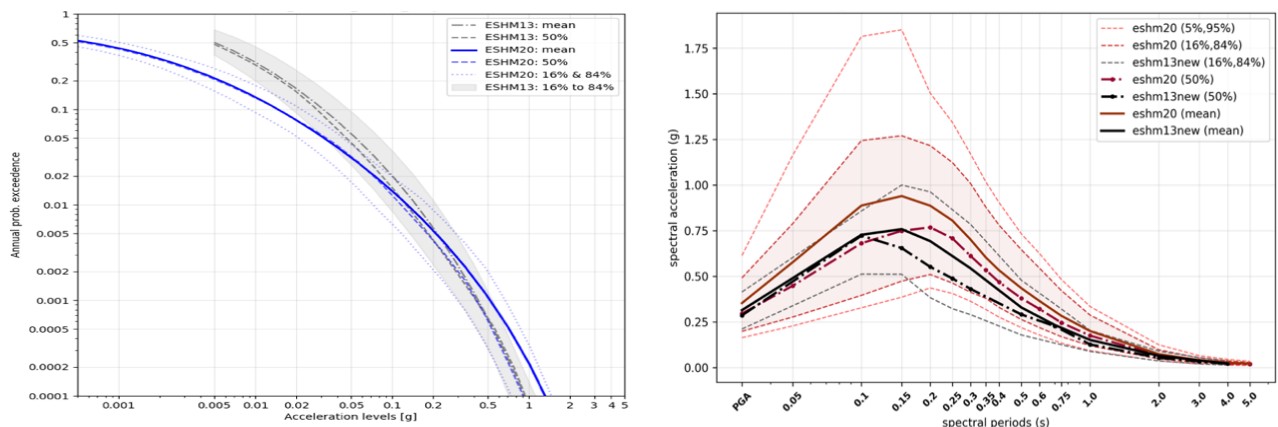


**Figure 6: Comparative plots of ESHM20 and ESHM13 for L'Aquila (42.3;13.382), Italy. The left panel illustrates the hazard curves for PGA distributions of ESHM20 and ESHM13. The right panel displays a comparison of the mean, median and the 5th, 16th, 84th and 95th quantiles of Uniform Hazard Spectra (UHS) for both models, for an APE 1/475. The shaded area highlights the 16th-84th interval. These results are applicable to generic rock soil conditions with a shear wave velocity (Vs30) of 800m/s.**

Figure 6 provides a comparison between the ESHM20 and ESHM13 for hazard curves (left panel) and for the UHS (right panel) for L'Aquila, Italy. The hazard curves cross each other at an APE of about 0.1, with the ESHM13 hazard curves being higher than the those of ESHM20 for lower acceleration levels, and then ESHM20 hazard curves exceeding those of ESHM13. The range between the two quantiles (16th and 84th) is also consistent. Next, the UHS comparison indicate that median spectral acceleration values are consistent with the previous model, ESHM13, for spectral periods less than 0.1 seconds. However, for

longer spectral periods, ESHM20 provides higher values. The mean and quantile values (5th, 16th, 84th, 95th) at APE 1/475 from ESHM20 are higher than those from ESHM13; on the contrary, the range between the 16th and 84th quantiles for L'Aquila of





ESHM13 is notably narrower than those in ESHM20. This difference is important because it reflects the methodological differences in the two regional hazard models. ESHM20's broader uncertainty range indicates potentially a more robust distribution around the mean and median values, a result of the balanced logic tree of both seismogenic source and ground

motion models. This is evident in Figure 6, where the mean and medians are depicted together with the distribution of hazard curves for each of the 10,000 samples of the logic tree. It is important to reiterate that the ESHM13 quantiles are sensitive to the model implementation, which did not explore the full logic tree of the source model and ground motion due to computational constrains. The observations for L'Aquila are not representative of the entire model, as at each site across the Euro-Mediterranean region, the hazard estimates are driven by different components of the model. Hence, to explore additional

materials and comparison plots of the ESHM20 and ESHM13, we encourage the readers to visit the dedicated repository listed in the "*Datasets and Online Resources*" section.

Next, in Figure 7 we illustrate the spatial distribution of the spectral acceleration at the fundamental period of vibration of 0.2s i.e., SA (0.2s) mean values for APE of 1/475. This hazard map incorporates both the uncertainties of the seismogenic sources and the spatially variable backbone ground motion model and uncertainty that is depicted by percentile estimates (5th,95th),

which are also shown in Figure 5. The distribution of equiprobable probabilistic ground motion levels is consistent with and resembles the patterns observed in regions with a history of documented seismic activity or located near active faults. High ground shaking regions are observed along the Mediterranean belt, extending through Greece, Italy and parts of the Balkans. These regions are tectonically active due to the complex interactions between the African, Eurasian and Anatolian tectonic plates. Similarly, in the Iberian Peninsula, Southern Spain and Portugal show a moderate to high level of ground shaking

hazard. This is consistent with the historical and geological data, as these regions have experienced significant earthquakes in the past, such as the 1755 Lisbon earthquake. Low ground shaking levels are visible in the northern, northwest and central parts of Europe, including the UK, northern France, Germany and Scandinavia. These regions exhibit low seismotectonic activity and historically experienced less frequent and severe seismic events. For the first time, the model includes the Canary Islands. The inclusion of the Azores Islands is also worth noticing, as this archipelago is another seismically active region.

This is due to the complex tectonic junction of three major plates (i.e., Northern America, Eurasia and Africa plates), resulting in elevated levels of seismic hazard. Likewise, the ground shaking values for Iceland fall within the moderate to high range, incorporating both shallow seismicity from tectonic origin and volcanic activity. This reflects the intricate nature of the Mid-Atlantic Ridge, as well as the relatively frequent volcanic activity. The hazard maps for the 16th, 50th and 84th quantiles are also illustrated in Figure 7. Last but not least, at the time of this submission, the disaggregation matrices of different ground

motion levels corresponding to the APE 1/475 and 1/2475 is scheduled to be uploaded on the online repository. The disaggregation of seismic hazard will provide insights of the predominant scenarios (magnitude and distance) at the selected AFE for all computational grid points of the Euro-Mediterranean region.





**Figure 7: Spatial distribution of the Spectral Acceleration at 0.2sec mean values for ESHM20 with an APE 1/475; the 16th, 50th and 84th quantiles are shown in the lower panel, all values are estimated on a generic rock soil conditions of $V_{s30}$~800m/s.**



### 6.2 ESHM20: Hazard Contribution per Seismogenic Source Models

Hazard maps for each component of the seismogenic source model, i.e., the area sources model and the hybrid source model
combining the active faults plus background smoothed seismicity, subduction in-slab plus deep seismicity as well subduction
interface are given in Figure 8. These hazard maps depict the mean PGA values for APE 1/475, for a reference rock condition
of Vs30~800m/s. The calculation is done with the corresponding logic tree of each source model (i.e. for the shallow crust,
the upper or lower part of the logic tree described in the upper panel of Figure 4) and the full backbone ground motion models.
The area sources, illustrated in Figure 8(a), provides a narrower range of PGA values, than the result computed with the hybrid
model. The delineation of the area source geometries is visible in the ground shaking map, in particular in regions of low to
moderate seismicity (i.e., Spain, France, Germany, UK and northern Europe). Moreover, as expected, the hybrid model shows
peaks of ground motion along the main seismically active faults incorporated in the model. The hybrid source model, given in
Figure 8b, provides the largest PGA values in regions with faults of high seismicity such us the North and East Anatolian Fault
Systems, Turkey, Corinth Gulf, Greece, or central Apennines in Italy. The contribution of background seismicity in the
proximity of faults results in similar patterns with the area source in regions like the Balkans, southern Spain, Portugal, or
France. The hazard pattern due to smoothed seismicity differ slightly from area sources in the central and northern Europe,
indicating the spatial variability of the adaptive kernel versus the more rigid approach based on area sources. The contribution
of deep seismicity to the overall hazard, as seen in Figure 8c, is typically lower than that the other source models. However,
the Vrancea source is an exception, exhibiting remarkably high hazard estimates with PGA values that are strongly azimuth-
dependent relative to the orientation of the Carpathian Mountains. As indicated in Figure 7d, the subduction interface model
is particularly important in regions prone to subduction zone earthquakes in southern Europe. This model shows a wide range
of PGA values depending on the specific characteristics of the subduction interface. Its contribution can be significant in terms
of potential ground shaking levels due to the large magnitudes associated with subduction zone sources.

It is important to note, that the subduction sources of Hellenic, Cyprian, Calabrian and Gibraltar Arcs, along with the deep
seismicity sources, complement the two source models proposed for the shallow crust: the area sources and the hybrid faults
and background smoothed seismicity. If we consider lower APEs i.e., 1/2500 or 1/5000, the relative contributions of each
component of the source models to the overall hazard are expected to change. The recurrence rates and higher magnitudes,
which play a more important role in controlling the hazard at these low return periods, will likely identify the contribution of
active faults and the subduction interface, due to the increased contribution of the $M_{max}$. In general, the components of the
source model make a balanced contribution to the overall hazard, as shown by the distribution of the hazard curves illustrated
in Figure 5. This is the outcome of the first seismogenic source model in Europe which accounts for both the spatial and
temporal variability of each component as well as the epistemic uncertainties of the individual seismogenic sources.



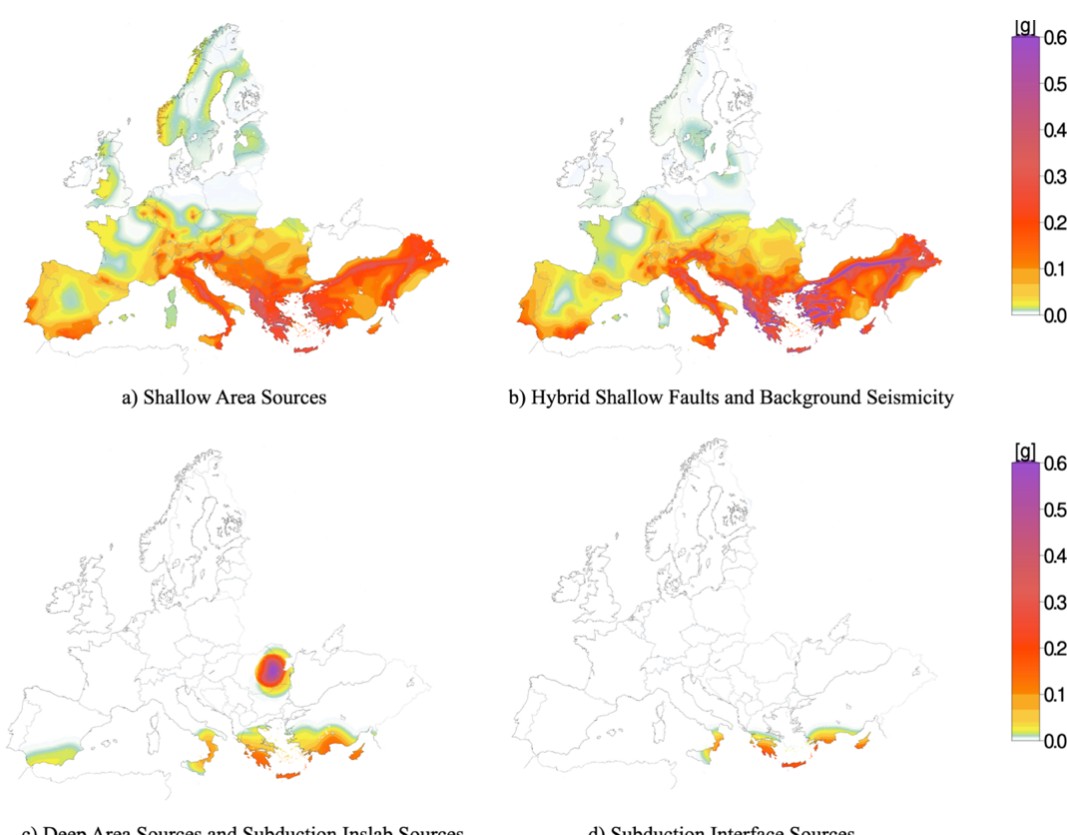

**Figure 8: Ground shaking hazard maps for PGA [g] mean for the main components of ESHM20 seismic source characterisation:(a) shallow area sources, (b) hybrid model combining the active faults and the smoothed seismicity, (c) deep seismicity and subduction intraslab and subduction interface (d) models for an APE 1/475. These results are applicable to generic rock soil conditions with a $V_{s30}$ of 800m/s.**

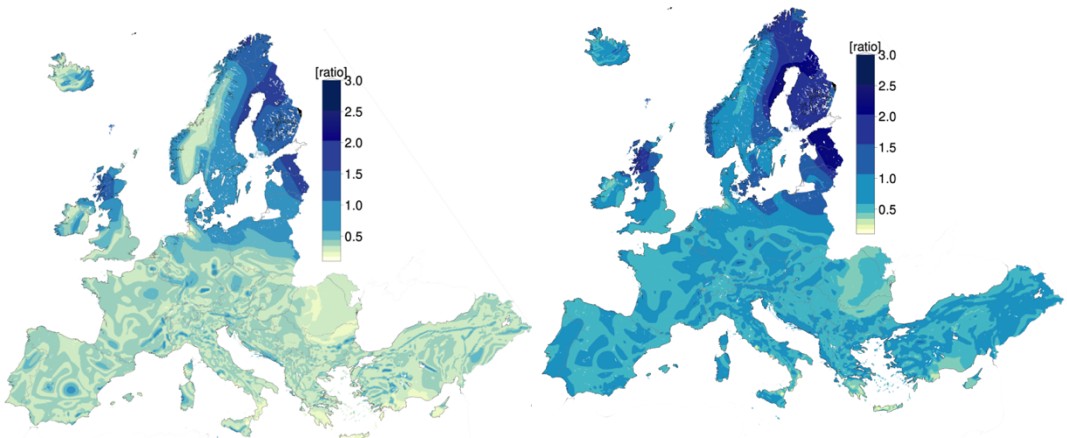

**Figure 9: Spatial distribution of quantiles ratios (84th to 16th) (left) and 95th to 5th (right) for ESHM20. The ratio depicts PGA for return period of 475 years and a reference rock condition of $V_{s30}$~800m/s.**



### 6.3 ESHM20: Epistemic Uncertainty Range

We use the quantile ratios i.e., $\log10(95^{th}/5^{th})$ and $\log10(84^{th}/16^{th})$ to map the spatial variability across the Euro-Mediterranean region. Figure 9 illustrates the spatial distribution of these two ratios for PGA at an APE of 1/475. The spatial pattern appears

to be similar between the two quantile ratios and as expected the $95^{th}$ to $5^{th}$ quantile ratio is generally higher than the $84^{th}$ to $16^{th}$ quantile ratio. Relatively high values i.e., a factor greater than 3 are shown in Northern Europe, the Baltic area and specific regions in northern Scotland. These high ratio values are mainly due to very low values of the $5^{th}$ and $16^{th}$ quantiles as seen in the hazard curves for Stockholm in Figure 5. Note that, the ratio values are in log10 scale, thus the verry low values of 0.1 will correspond to a very narrow range of about 1.25 between the quantiles (mainly the $84^{th}/16^{th}$), which might suggest a very

strong correlation of the logic tree branches, resulting in overlapping end-branches. However, in the majority of regions the $84^{th}/16^{th}$ value is lower than 1.0, resulting in a range of up to 10 between the $16^{th}$ and $84^{th}$ PGA values. This range increases in the northern Europe, where the $84^{th}/16^{th}$ ratio values are greater than 1.5, resulting in a very high range between these two quantiles for the estimated PGA. Similarly, the spatial pattern of $95^{th}/5^{th}$ depicts very high value in the Northern Europe, suggesting a very large variability of the ground motion hazard estimates in the region. This observation is consistent with

what was found in ESHM13, albeit the range of ratio values is slightly lower, mainly due to different logic tree configuration and model implementation (Danciu et al., 2022). However, the quantile ratios exhibit values that are in line with other regional seismic hazard models (Şeşetyan et al., 2018) or site specific seismic hazard models (Douglas et al., 2014).

### 6.4 ESHM20: Transfer to Engineering Community

The engagement of the scientific and engineering communities has been a priority in the development cycle of the ESHM20

since the early stages of the project. The engineering requirements were specified by the CEN/SC8 working group, which is coordinating the definition of the seismic action of the next version of the European seismic design code (Bisch, 2018). Their requirements were aligned with the new definition of the standard response spectra, particularly with regard to its anchoring points. Specifically, the current scaling practice based on a single parameter (PGA), is being replaced by a new anchoring system with two parameters: $S\alpha$, which is the spectrum value on a plateau that covers a certain spectral period range (i.e., 0.05s

and 0.4s) and $S\beta$, which is the spectrum value at 1s spectral period (Labbé and Paolucci, 2022). The pseudo-acceleration response spectrum continues to be the basis of the new design spectra definition in EC-8. The ESHM20 core team has interacted with the SC8 working group for knowledge-transfer, providing access to all inputs, datasets, models and results. Several bilateral meetings and four plenary meetings took place in the last phase of the model development cycle from 2020 to 2021. Moreover, the ESHM20 was evaluated with different comparison and testing activities for several countries in Europe: France,

Greece, Italy, Norway, Portugal, Romania, Slovenia and Switzerland. Iervolino et al., 2023, has analysed two Italian seismic hazard models (MPS04, Stucchi et al., 2011 and MPS19, Meletti et al., 2021) and ESHM20 against ground motion recordings. The strong motion data was gathered from 143 seismic stations of the Italian seismic network, and the analysis was conducted for several intensity measure types i.e., PGA, SA at 0.3s and SA at 1s, as well as four return periods. A hypothesis test was



setup to account for the number of exceedances observed at each station, and if a value falls within the non-rejection band, the

test is considered successful. In total, thirty-six hypothesis tests were performed, each representing the combination of all

models and parameters. The results suggested that all three models (i.e., MPS04, MPS19 and ESHM20) perform adequately

considering a 5% significance level. Another important finding of the study was that the all-hazard models tend to overpredict

exceedances at 50-year return period.

The two informative hazard maps, namely, $S\alpha,475$ and $S\beta,475$, as derived from the ESHM20 for a 475-year return period, for

soil type A (Vs30~800m/s) as requested by the CEN/EC8 are shown in Figure 10. To calculate the $S\alpha475$, it is first necessary

to identify the $T_{peak}$ which corresponds to the largest value of the median UHS for a 475-year return period; then the $S\alpha475$ is

estimated as an average of spectral values between the periods range of $0.5T_{peak}$ to $1.5T_{peak}$. The use of median values was a

decision taken by the CEN/TC 250/SC8 and summarized in Labbé and Paolucci, 2022. Additionally, the CEN/TC 250/SC8

states that while these median values are used for these specific informative maps, they do not pass judgment on other results

from the ESHM20, maintaining a neutral view of hazard values other intensity measure types or other return periods (Labbé

and Paolucci, 2022). These informative hazard maps are now subject to the inquiry and formal voting process conducted by

CEN member nations However, it is also critical to emphasize that the results obtained from ESHM20 should not be directly

used as seismic design values. Instead, seismic design provisions and national annexes should be followed and enforced for

seismic design purposes.

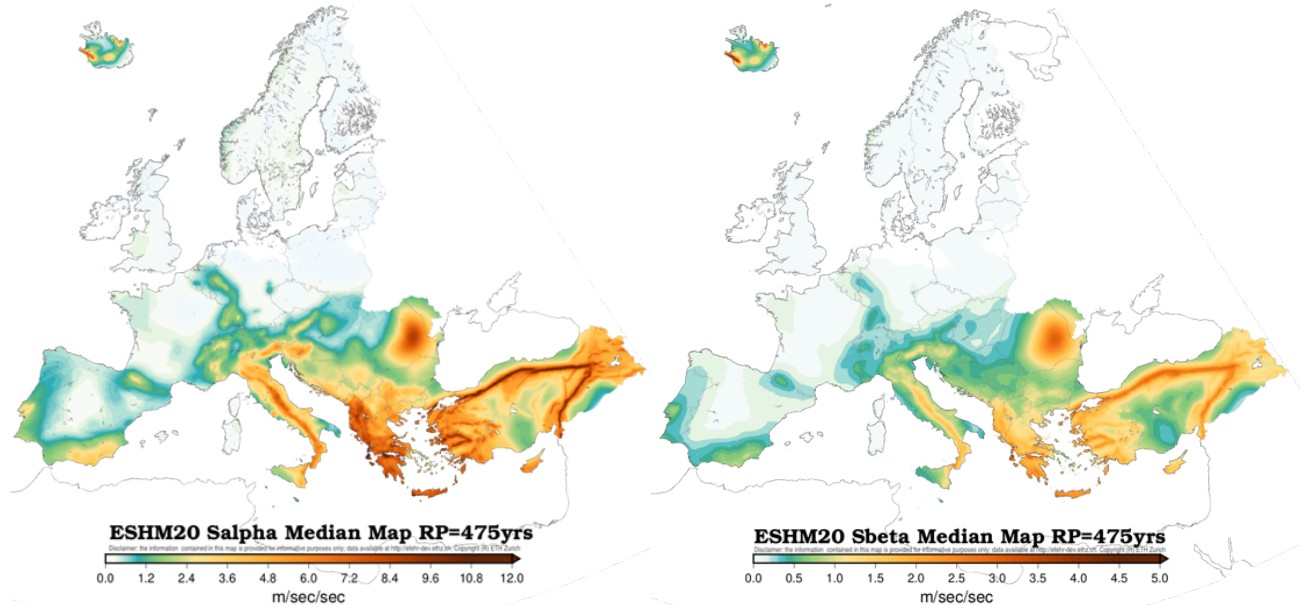


**Figure 10: Informative hazard maps for median $S\alpha,475$ (left) and $S\beta,475$ (right) values for 475-year return period and class type A of EC8 with a $V_{s30}$ of 800m/s (Labbé and Paolucci, 2022).**



## 7 Summary and Outlook

The European Seismic Hazard Model (ESHM20) represents a significant step forward in the assessment of seismic hazard for the Euro-Mediterranean region. It incorporates the latest data sets, as well as advanced methods for building the model and computing hazard when compared to ESHM13. ESHM20 was developed through a collaborative, regional effort that integrates the expertise and data from various European countries. The development of ESHM20 involved a rigorous process of compilation, standardization and curation of input data, including earthquake catalogues, ground motion recordings and

models, tectonic information, active faults databases, and/or seismogenic sources from various countries. A key feature of the model is its cross-border harmonization, which guarantees a consistent assessment of seismic hazard across national borders. ESHM20 supersedes ESHM13 as a reference model for seismic hazard assessment in the Euro-Mediterranean region and it is also part of the recent update of the GEM Mosaic of Hazard Models (Pagani et al., 2020).

From the communication point of view, an important milestone was achieved with the first-ever release of a European regional

seismic hazard and risk model to the scientific community and the general public. In May 2022, an official media release was made, accompanied by various materials such as flyers, fact sheets, posters, and video content. These materials were translated into multiple languages, facilitating the dissemination of the model's findings to a wider audience. This media event played a key role in enhancing awareness and understanding of earthquake hazard and risk in the Euro-Mediterranean region (Dallo et al. 2023).

EHSM20 results show differences from the results of national seismic hazard models, mainly due to variations in datasets and methodologies used (Šket Motnikar et al., 2022; Grünthal et al., 2018; Wiemer et al., 2016; Mosca et al., 2022; Vacareanu et al., 2016; Meletti et al., 2021; Halldórsson et al., 2022). These differences should be explored and understood to fully integrate the insights of the regional models into a national hazard (Pavel et al., 2016; Weatherill et al., 2023b; Trevlopoulos et al., 2023) or risk models (Papadopoulos et al., 2023). Note, that albeit of key importance, there is no straightforward way of

understanding the importance and the significance of the changes between seismic hazard models. What may be considered low seismic activity in specific region or country (e.g., southern Europe) might be classified as high in another specific region (e.g., northern Europe), and vice-versa. The impact of earthquakes depends on factors such as building vulnerability, local site conditions, asset exposure and population density. Thus, to understand and compare the impact of earthquakes in different regions, the ground-shaking estimates must be analysed in the context of seismic risk  (ESRM20, Crowley et al., 2021).

Furthermore, the availability of products such as ESHM20 and ESRM20 opens the possibility of developing operational services in Europe, such as applications similar to the ShakeMap service, time-dependent Earthquake Forecasts or Rapid Loss Assessment. Some of these applications have been developed or are currently being developed in recent pan-European initiatives (Spassiani et al., 2023; Mancini and Marzocchi, 2023; Böse et al., 2023).

Despite its advancements, ESHM20 faces challenges typical of large-scale seismic hazard models (Gerstenberger et al., 2020).

The heterogeneity of data used and their variability in completeness, particularly in regions of low seismicity, pose a significant challenge to the development of a uniform hazard model. The seismogenic source model, while comprehensive, may require





future refinements as new data and models are integrated, particularly geodetic and geologic data. However, in the recent years significant progress was made in regional seismic hazard models, including those in the US (Petersen et al., 2020; Field et al., 2017), New Zealand (Gerstenberger et al., 2022), Australia (Allen et al., 2020), Italy (Meletti et al., 2021), Germany (Grünthal et al., 2018), Switzerland (Wiemer et al., 2016)and at a global scale (Pagani et al., 2020) , it is worth observing that the overall uncertainties have not seen a significant reduction over time. The complexity of the seismic hazard models steadily increased given the continuous expansion of earthquake data and information, leading also to an increase in uncertainties around these models that previously neglected. Looking ahead, one can anticipate the development of more advanced methods for handling the uncertainties in seismic hazard datasets, components and models. These methods are likely to involve the use of physics-based simulations of both earthquake ruptures and/or ground shaking (Bradley, 2019; Paolucci et al., 2021; Li et al., 2023). The integration of physics-based simulations into seismic hazard modelling has the potential to improve the main pool of existing records, enhance ground shaking characterization particularly in the magnitude-distance range poorly covered in current strong-motion databases, improve the ground motion characteristic models, augment our understanding of earthquake scenarios, and support earthquake preparedness and mitigation strategies (Graves et al., 2011). In addition to using physics-based simulations, one can also anticipate significant progress in computational capabilities, such as high-performance computing (Folch et al., 2023), and artificial intelligence (AI)-driven analytics in the coming years. These leading-edge tools have the potential to significantly accelerate the processing of extensive, multidisciplinary datasets and complex calculations (Dal Zilio et al., 2023). However, the future application of AI and machine learning in seismic hazard modelling is not without challenges, including black box dilemma, computational demand, scalability, representation of data, bias, over fitting, ethical considerations (Jiao and Alavi, 2020; OECD, 2023). Furthermore, it is evident that we are entering a decade of interoperability and close interdisciplinary collaboration. Advanced research infrastructures such as the European Plate Observing System (EPOS, Haslinger et al., 2022), with their next generation of Earth science data and services, can facilitate such data interoperability for advancement in the field and support the next generation of geo-related hazard and risk models.

In conclusion, ESHM20 is foreseen as a living and collaborative model, that includes a broader European community of Earth scientists, seismic hazard experts and modellers, engineers, practitioners, and stakeholders. ESHM20 will coexist with the national hazard models, which is not intended to replace, but rather advance and enhance; this coexistence will rely on an ongoing feedback loop between the two, where new data, methodologies and/or innovative approaches will contribute to the future versions of these hazard and risk models. Ultimately, this will support efforts to create a more earthquake-resilient society in the Euro-Mediterranean region.



## 8 ESHM20: Datasets and Online Resources

**Documentation and Supplementary Files**

ESHM20 Technical Report:

https://gitlab.seismo.ethz.ch/efehr/eshm20/-/blob/master/documentation/EFEHR_TR001_ESHM20.pdf

- ESHM20 vs ESHM13 comparison hazard curves and UHS, for more than 400 sites (cities with a population larger than 100k): https://gitlab.seismo.ethz.ch/efehr/eshm20/-/tree/master/additional_materials

- Additional Plots, depicting the attributes of the individual source parameters, including spatial distribution of the activity
parameters (aGR, bGR, Mmax), hypocentral depth distribution, dip, rake, strike, upper and lower seismogenic depth:https://gitlab.seismo.ethz.ch/efehr/eshm20/tree/master/additional_materials/source_model_plots/attribute_plots

**Main Repository**

- ESHM20 main repository: https://gitlab.seismo.ethz.ch/efehr/eshm20

- An overview of the ESHM2020 project, complete with links to the main elements is available from
http://hazard.efehr.org/en/Documentation/specific-hazard-models/europe/eshm2020-overview/

**Communication and Outreach:**

- Posters, factsheets and additional media products: http://www.efehr.org/explore/Downloads-information-material/

- Web-page and Data Access: http://hazard.efehr.org/en/home/

- EFEHR web service, available at http://hazard.efehr.org/en/web-services/

- Specific web service links follow:
  - Hazard Curve data: http://hazard.efehr.org/en/web-services/hazard-curve-data/
  - Hazard Map data: http://hazard.efehr.org/en/web-services/hazard-map-data/
  - Uniform hazard spectra data: http://hazard.efehr.org/en/web-services/uniform-hazard-spectra/

**Author Contribution**

LD conceptualization, data preparation, methodology, formal analysis, software, visualization, writing the original draft manuscript. DG: conceptualization, methodology, coordination, and review. WG: conceptualization, data preparation, methodology, formal analysis, software, visualization, review and editing. RB: conceptualization, data preparation, methodology, formal analysis, software, visualization, review and editing. AR: data preparation, methodology, formal analysis, review and editing. CB: methodology, formal analysis, review and editing. PYB: methodology, formal analysis, review and
editing. MP: conceptualization, methodology, software, review and editing. SN: data preparation, methodology, formal analysis, software, visualization, and review., CR: data preparation, methodology, formal analysis, software, visualization, review, KS: methodology, formal analysis, and review. SV: methodology, formal analysis, and review. FC: conceptualization, methodology, coordination, formal analysis, review and editing. SW: conceptualization, methodology, coordination, formal analysis, and review.





**Competing interests:**

LD, DG, PYB, FC and SW are part of the editorial committee for this special issue of Natural Hazards and Earth System Sciences. Consequently, they are not eligible to serve as editors in charge of this manuscript.

**Acknowledgements**

The ESHM20 was developed within an open and collaborative framework of the EU-funded project "Seismology and Earthquake Engineering Research Infrastructure Alliance for Europe" (SERA Project, 2017-2020, Giardini et al 2020), as part of the INFRAIA-01-2016-2017 Research Infrastructure for Earthquake Hazard in Europe. ESHM20 was finalized in December 2021 and publicly released in April 2022.

We are grateful all researchers, scientist, experts and engineers contributing to the development of the 2020 European Seismic Hazard Model (ESHM20), in various way from data compilation and curation to knowledge transfer during numerous meetings and webinars: Andrea Antonucci, António Araújo Correia, Sinan Akkar, Kuvvet Atakan, Jure Atanackov, Hamid Sadegh Azar, Stephane Baize, Julien Barriere, Paolo Bazzuro, Maria Belen Benito, Myriam Belvaux, Bjarni Bessason, Dino Bindi, Christian Bosse, Thierry Cameelbeck, Eduardo Cansado Carvalho, Michele Carafa, Fernando Carrilho, Alexandra Carvalho, Carlo

Virgilio Cauzzi, Gilles Celli, Eugenio Chioccarelli, Carmen Cioflan, Pasquale Cito, Cécile Cornou, Edward Cushing, Susana Custódio, Snjezana Cvijic-Amulic, Nicola D'Agostino, Vera D'Amico, Nicolas D'Oreye, Atefe Darzi, Mathias Dolsek, John Douglas, Branko Dragicevic, Dejan Dragojević, Stephane Drouet, Blaise Duvernay, João Estevão, Donat Fäh, Ekkehard Fehling, Joao Fonseca, Mariano Garcia-Fernandez, Eulalia Gracia, Gottfried Grünthal, Benedikt Halldorsson, Florian Haslinger, Marian Herak, Jomard Herve, Stefan Hiemer, Fabrice Hollander, Shible Hussein, Iunio Iervolinio, Diethelm Kaiser,

Vanja Kastelic, Amir M. Kaynia, Rexhep Koci, Annakaisa Korja, Sreeram Kotha, Svetlana Kovacevic, Milad Kowsari, Olga-Joan Ktenidou, Daniela Kuehn, Kresimir Kuk, Neki Kuka, Pierre Labbe, Steffi Lammers, Giovani Lanzano, Björn Lund, Lucia Luzi, Francesco Maesano, Elena Manea, Päivi Mäntyniemi, Basil Margaris, Christophe Martin, Warner Marzocchi, Frederic Masson, Luís Matias, Carlo Meletti, Nikolaos Melis, Albero Michelini, Jadranka Mihaljevic, Zoran Milutinovic, José Antonio Peláez Montilla, Ilaria Mosca, Roger Musson, Shemsi Mustafa, Adrien Oth, Bruno Pace, Roberto Paolucci, Christos

Papaioannou, Florin Pavel, Laura Peruzza, Rui Pinho, Kyriazis Pitilakis, Valerio Poggi, Mircea Radulian, Evi Riga, Philippe Roth, Agathe Roulle, Zafeiria Roumelioti, Radmila Salic, Abdullah Sandikkaya, Antoine Schlupp, Jochen Schwarz, Anselm Smolka, Oona Scotti, Barbara Sket-Motnikar, Vitor Silva, Anne Socquet, Efthimios Sokos, Mathilde Sorensen, Carlos Sousa Oliveira, Thomas Spies, Massimiliano Stucchi, Nikolaos Theodoulidis, Mara Tiberti, Paola Traversa, Nino Tsereteli, Radu Vacareanu, Dina Vales, Roberto Vallone, Dimitrios Vamvatsikos, Kris Vanneste, Emmanuel Viallet, Daniele Vigano,

Francesco Visini, Stefan Weginger, Polona Zupancic



We would like to thank IT team at Swiss Seismological Service, ETH Zurich for their excellent support on maintaining the servers, update the databases and upgrade the EFEHR webpage and the *hazard.EFEHR.org* web-platform to access the 2020 models: Philipp Kästli, Leandra Eberle, Emil Zylis, Emilia Petronio and Cyrill Bonjour.


Furthermore, we would like to also thank Helen Crowley and Jamal al Dabeek for their efforts and feedback regarding the use of the seismic hazard for risk calculation. We would like to extend our gratitude to Michèle Marti, Nadja Valenzuela, Irina Dallo and Simone Zaugg for their creativity and coordination of the outreach activities. In addition, we would like to thank Michele Simionato for timely OpenQuake Engine improvements.


Finally, the authors would wish to express their appreciation to the SC8 Ad-hoc Working Groups for the excellent collaboration during the ESHM20 evaluation: Pierre Labbe, Philippe Bisch, António Araújo Correia, Hamid Sadegh Azar, Alexandra Carvalho, Matjaž Dolšek, Blaise Duvernay, João Estevão, Ekkehard Fehling, Max Gündel, Iunio Iervolino, Christophe Martin, Roberto Paolucci, Evangelia Peli, Kyriasis Pitilakis, Jochen Schwarz, Mathilde Sorensen, Evi Riga, Kris Vanneste, Dimitrios

Vamvatsikos, Emanuel Viallet, Radu Vacareanu.

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
