# Peer review of "The 2020 European Seismic Hazard Model: Overview and Results"

_EGUsphere, 2023_

## Author Comment (AC1)

**Reviewer 1**

The manuscript by Danciu et al. provides a concise and short description of the European Seismic Hazard Model 2020. The overview paper lists, describes and references the vast amount of work that build the foundation the ESHM20 and will serve as primary reference for this model, pointing to more detailed descriptions of the ingredients of the model. This manuscript is a significant contribution to improve seismic hazard assessment efforts on the regional European scale.

I recommend publication of this manuscript after minor revision. I primarily request clarifications and clearer statements of the authors opinion about some of the results. Please guide the reader with your interpretation of the results rather than let the reader free to do so.

We would like to express our gratitude to the reviewer for taking the time to review our manuscript. We greatly appreciate your constructive comments and the opportunity to improve our work. We value your suggestions and have addressed them sequentially.

Below I have listed my requests pointing to the lines in the manuscript.

Lines 117-118: Please make the link to the additional data more prominent. It took me a while to figure out that I have to go to section 8 to access supplementary data. Add a link here and reference the section, a table, something that is easy for the reader.

Reply: We will address this point in the revised manuscript (i.e., a link and reference to be added). Note that the structure of the manuscript follows the journal format, with a **Data Availability** section at the end of the manuscript.

Line 143: An example would be handy and would give real numbers for the reader: What is the overall rate change for M≥6 (or some threshold) in the model (overall). What is the change in active vs stable regions? Please add an MFD rate comparison plot similar as in Woessner et al. (2015) for the source models and the weighted combination.

Reply: The actual rate forecats of the M>= 6.4 are given in Section 4,starting with line 307. A corresponding source model MFD similar with Woessner et al 2015, will be added. The differences in the net yearly rate between the two models are given in the title of the difference rate map in Figure 1 of Supplementary Materials. The main manuscript will provide these values for the reader's reference.

Lines 154-155: Isn't this (at least partly) mitigated by the regional adjustments of the non-ergodic GMM?

Reply: yes, that is a very good observation. The text will be changed to include the fact that the GMM regionalisation, can be considered as a partially non-ergodic model but not fully ergodic.

Lines 150-162: In this section, a discussion on the regional impact would be good. PGA475 increases in Romania strongly, increases are also seen in Spain. Decreases are seen in Italy and Turkey. Can you point out spatially (by tectonics, by country) the main drivers of change: Is it the rate model and if so which part of the rate model?  Is it the weighting or the models?  Is it the ground motion model, which decision therein?  Is the ground motion model update more impactful than the update of the source model? Are there spatial differences?

Reply: The aim of this section is not to depict in great detail the regional variability of these differences. The earthquake rate maps and trellis plots are meant to represent the main driving factors of these differences. It is beyond the scope of this manuscript to depict every single aspect of the differences at the country level. The compponents of both components hazard models are openly available, However, we might consider to follow-up with an additional manuscript to describe in greater details these differences.

Lines 212-214: Could you elaborate on these tests and the criteria used to decide on a declustering method? More precisely, was the catalogue (against which the model was evaluated) declustered as well with the same method? If no, how insightful is a comparison of a mainshock forecast with a full (foreshock, mainshock, aftershock) catalogue? Sentence is not clear on which method was used in the end. Please clarify.

Reply: The default declustering method of the unified earthquake catalogue is based on the method proposed by Grünthal, 1985, the same one used in ESHM13. This was a decision made at the beginning of the project.

In the statistical testing framework, we applied Kolmogorov-Smirnov (KS) tests to the time distribution of all resulting mainshock catalogs, revealing that all declustering techniques produced catalogs conforming to a stationary Poisson distribution across most tectonic regions. Due to the indistinguishable outcomes from the KS tests among the declustering methods, we decided to avoid choosing a method based solely on these results. Instead, we shifted our approach to utilize the Collaborative for the Study of Earthquake Predictability (CSEP) comparison likelihood tests, which evaluate the performance of declustering by using the same methodology for earthquake activity rate estimation while varying the input catalog according to the declustering method.

For the catalogs, we constructed activity rate forecasts using data from declustered catalogs up to December 2006, referred to as the "learning catalog". Activity rates were estimated for each completeness super zone, employing the Weichert (1980) method to compute a- and b-values for the Gutenberg-Richter relationship. The "target catalog" consisted of all events above moment magnitude Mw 4.5 from January 2007 to December 2014, containing 2107 events. This served as the benchmark for our forecasting models' performance evaluation.

To determine the model with the best forecasts , we utilized CSEP's comparison likelihood tests, namely the T-test and W-test, designed to measure the rate-corrected average information gain per earthquake and determine if one model's forecast skill significantly exceeds that of another. Our approach ensures consistency by using the same declustering method for both the learning and target catalogs, providing a solid basis for our comparisons.  Thus, we used the same declustering method for both the learning and target catalogs to ensure a consistent basis for our comparisons. In conclusion, we conducted the statistical analysis as a sensitivity test to assess the influence of the declustering techniques on earthquake rate forecasts. In the manuscript, we will explicitly state that the Grünthal (1985) method is the standard approach employed for declustering. We hope this response clarifies the use of the declustering methods in the model development. The manuscript will be modified accordingly.

**Additional references:**
Geophysical Research Abstracts, Vol. 21, EGU2019-13126, 2019
https://meetingorganizer.copernicus.org/EGU2019/EGU2019-13126.pdf
SERA Deliverable 24.4:

http://static.seismo.ethz.ch/SERA/JRA/SERA_D24.4_Test-bed%20validation%20of%20tools%20and%20resulting%20high%20level%20products-%20software%20toolbox,%20validation%20methodologies,%20demonstration%20report.pdf

Line 216: "this branch" · which branch?

Reply: Logic tree branch corresponds to the Reasenberg 1985 declustering. Text will be adjusted accordingly.

Lines 264-265: "as a proxy of the spatial organization of the source models" · what does this mean?

Reply: In the sentence, "The TECTO zonation layer, as shown in Figure 2b, is used as a proxy for the spatial organization of the source models and provides a basis for estimating more stable Gutenberg-Richter (GR) parameters given the larger number of earthquakes within each super zone," we refer to the following:
-   The TECTO zonation layer describes how various groups of seismic sources are distributed spatially. By "spatial organization of the source models," we mean the arrangement and classification of seismic sources in geographical space as they are represented within our model. The TECTO layer encapsulates this by delineating areas with similar seismic characteristics, which are inferred from geological, geophysical, and seismological data.
-   In the Tecto layer, each polygon corresponds to the combination of several area sourcesThe set of earthquakes within each TECTO zone is larger than the one for each source. As a result, the parameters of the GR relationship in each TECTO zone are statistically more robust than the one for each zone (but less representative of possible local variations).

In essence, the TECTO zonation layer aids in the structuring of our seismic hazard model by providing a clear, organized basis upon which seismicity can be analyzed and interpreted. This organization is key for the accurate estimation of seismic hazard parameters, ensuring that our model reflects the geographical variation in earthquake processes across different regions. We hope this explanation clarifies the intended meaning of the sentence and the role of the TECTO zonation layer in ESHM20.

Lines 274-276: How exactly were the three Mmax values selected? Do they correspond to specific quantiles of area?

Reply: Yes, the Mmax values for each fault were chosen to represent the 2%, median, and 98% quantiles of the Mmax range, which depicts the variability and uncertainty in fault parameters (considering all combinations of length, width, aspect ratio, area considering the dip angle variability) and magnitude scaling laws. The text will be changed accordingly, also adding a reference to Basili et al. (2023) (already cited), where this procedure is explained in more detail.

Line 278: "lower value of the expected Mmax" · maybe remove "expected"

Reply: to be corrected accordingly.

Lines 319-320: Are "similar predictions for well-represented scenarios" not desirable? Please clarify.

Reply: We are specifically referring to "well-represented" scenarios, as those magnitude-distance ranges for which the strong motion data is abundant, i.e. magnitude range 4.5 to 5.5, and distance 30 to 100 km. Text adjustment will be done for clarity.

Lines 367-369: Some of the uncertainties explicitly explored in ESHM20 were "collapsed" In ESHM13 (e.g. b values). This sentence suggests that ESHM13 featured a single collapsed source logic tree branch, although in reality it had three alternative branches representing different methodologies. Please clarify.

Reply: That is a very good observation. The text will be improved for clarity: "..*a collapsed earthquake rate was used for each of the three seismogenic source models.*"

Lines 383-385: What sensitivity studies were conducted, and what criteria were considered to decide on using the "correlated end-branches"?

Our sensitivity analyses assessed the impacts of using correlated versus uncorrelated activity rate parameters i.e., aGR, bGR-parameters, and Mmax.  A new reference is added to the document:

*Papadopoulos, A. N. and Danciu, L. (2022): A critical look into the discretization of epistemic variables and treatment of their correlation in seismic hazard and risk assessment, in: Proceedings of the Third European Conference on Earthquake Engineering and Seismology – 3ECEES, 3rd European Conference on Earthquake Engineering and Seismology (3ECEES 2022), Accepted: 2023-10-23T09:01:45Z, 3849–3856, https://doi.org/10.3929/ethz-b-000593649, 2022.*

The primary criteria, as explained in the manuscript is the balance between computational manageability and the meaningful representation of seismic hazards. In conclusion, with today's hardware and software, implementing uncorrelated uncertainties for probabilistic ground shaking calculations at a regional level, such as in the as in the Euro-Mediteranean region, is impossible.

Line 415-418: This statement is vague. Please outline what would be required to be able to trust seismic hazard values at APE of 1/5000 or lower if required for a certain location and give references to example studies?

Reply: We acknowledge the term "trust" can be subjective; however, our intent is to highlight the challenges inherent in extending regional seismic hazard assessments to very low APE levels, which are traditionally better suited for site-specific hazard analyses following standards like the SSHAC guidelines.
For seismic hazard assessments at these very low probabilities,  site-specific site response analysis are usually performed (including a detailed evaluation of the associated epistemic uncertainty). Additionally, an extended observation period for earthquake sources, along with comprehensive geologic and geodetic data, becomes increasingly important to accurately capture the seismic hazard. It should be noted that the current ESHM20 model, similar to its predecessor ESHM13, is not specifically calibrated for APE values lower than $10^{-4}$ due to such complexities, as well as the completeness of the main datasets for such low probabilities. As a result, while ESHM20 provides valuable regional insights, for very low probabilities, we strongly recommend exploring the low probabilities with caution, and when applicable, a site-specific approach should be preferred. We will revise our manuscript to clarify these points and enumerate some of the limitations in interpreting hazard values at such low APE levels.

Line 430-444: Selecting these six locations to illustrate the complexity of the hazard curves and its uncertainty is illustrative. Despite this being a combined effect, could you elaborate in addition to the current description whether the uncertainty of the ground motion model or the rate model is more influential and for which location? Towards the end, the hazard curve 'decay' is described. The

description does not discuss the parameter that is responsible which I believe is the uncertainty of the GMPEs? Please clarify and describe more precisely.

Reply: Indeed, Figure 5 aims to showcase the combined effects that contribute to the seismic hazard at each site, emphasizing the overall uncertainty that our models capture. The full distribution of the logic tree branches represents the epistemic uncertainty, which significantly influences both the shape and range of the hazard curves. This encompasses uncertainties from both the ground motion models (GMMs) and the seismic rate models. It is important to note, however, that isolating the impact of GMM uncertainty versus seismic rate model uncertainty for each location presents substantial challenges. The interaction between these uncertainties varies across different geographic and seismotectonic settings, making it difficult to definitively attribute the influence to one component over the other at each specific location. Thus, we acknowledge that a detailed, location-specific description of these factors afecting the distribution of the hazard curves is beyond the aim of the current manuscript.

Furthermore, regarding the 'decay' observed in the hazard curves, this aspect is, a priori, primarily influenced by the b-values, activity rates, maximum magnitude, and the aleatory variability of the GMPEs. We have conducted a specific sensitivity analysis of the decay rate, expressed either as the ratio between hazard estimates at two different return periods or as the corresponding hazard curve exponent k: $SA(RP_2) = SA(RP_{ref}).(RP_2/RP_{ref})^{1/k}$, where $RP_{ref}$ is the "reference" return period (e.g., 475 years, and $RP_2$ is another, usually longer, return period (i.e., 2475 or 5000 years). The results have been presented at 3ECEES (Bard et al., 2022, slides available upon request) and will be presented in a forthcoming paper (Bard et al., 2024). They can be summarized as follows:

· While the default k value considered in many building codes, including EC8, is 3, the average mean value between 5000 and 475 years for ESHM20 results across all considered geographical areas, ranges between 1.6 and 1.85 whatever the spectral period (from PGA to 5 s)

· This exponent value is strikingly increasing with increasing hazard: it is below 1.4 in northern craton areas, below 2 in the northern and western moderate seismicity areas (e.g., England, most of Germany, Netherlands, Belgium and France, expect the more active areas such the Rhine valley and graben, western Alps and Pyrénées, and generally below 2.5 for most shallow active crustal areas

· Surprisingly, it does not exhibit any clear trend with $b_{GR}$, it is only very slightly increasing with increasing $M_{max}$, and it is NOT primarily driven by the ground motion aleatory variability : tests with a zero aleatory variability indicate it increases the hazard exponent mainly in high seismicity areas and only for very long return periods (beyond 5000 year) for low to moderate seismicity areas of northern and northwestern Europe. The hazard e-curve exponent may be also locally decreased close to active faults identified in moderately active zones where the long return period hazard estimates are dominated by fault activity.

Additional references:

Bard, P.-Y., L. Danciu & C. Beauval, 2022. Hazard curves, spectral shapes, importance coefficients and return periods: insight from recent PSHA studies. Invited theme lecture, Third European Conference on earthquake Engineering and Seismology, Bucharest, September 4-9, 2022.
Bard, P.-Y., B. Derras, C. Beauval, and L. Danciu, 2024. Hazard curve exponents and importance factors for EC8-like codes: a statistical analysis of ESHM20 results throughout Europe. In preparation.

Lines 458-460: Maybe also the fact that ESHM20 uses "correlated end-branches" which will overestimate the uncertainty, whereas ESHM13 only considers mean recurrence parameters (which underestimates uncertainty)?

Reply: That is indeed the outcome of the implementation of the logic trees of the models, i.e., full logic tree of the ESHM20 vs collapsed rate forecast for ESHM13. Between the two, the ESHM20 implementation is a state-of-practice implementation resulting in direct estimation of the result's quantiles from a full logic tree sampling; whereas the quantiles of ESHM13 are estimated in postprocessing and are based on a reduced number of logic tree branches. If needed, we will clarify this aspect further in the manuscript.

Figure 5: Please add a table with values to make comparisons easier or provide in the appendix. Including a table with these values for each capital city would be handy for reference.

Reply: All hazard estimates and products of ESHM20 and ESHM13 are available online at hazard.efehr.org. The reader can retrieve all the values and perform comparisons on their own at any location. A reference table with the hazard values of the capital cities will be provided in the Supplementary Materials.

Line 477 - 485: For the Azores, Canaries and Iceland: The Canaries are tectonically different from the Azores and Iceland. Has there been any differentiation in approaching this in the model?

Reply: No, the Canaries were treated as active shallow crust regions.

Line 485: Please add statements on whether differences for SA=0.2s and 1s are similar or different from what is seen for PGA. If not, please give examples for locations where differences occur and explain.

Reply: For specific APEs, there will be differences in the spatial pattern as well as the SA values at 0.2s, 1s, and PGA. Such differences can be spatially due to inherent differences between these factors (depicting the peak ground shaking, i.e., PGA, high frequency, i.e.,. SA0.2s or long periods SA1s) and their correlation with the source parameters, i.e., magnitude, different path attenuation, or site properties. We encourage the reader to explore the additional hazard maps available online at hazard.efehr.org.

Lines 506 - 507: The statement that PGA hazard patterns 'differ slightly' between the area and the smoothed seismicity model is not correct. Firstly, how do you define slightly? Secondly, there are differences in pattern and levels of PGA in Norway, Sweden, the UK, Belgium and Belgium - France border, German-Czech border, Southern Germany. It is positive that there are differences, otherwise two branches would not be needed. Please note and state these differences. Clarify what is considered different with a quantifiable metric? What does this tell the reader about the level of knowledge in a certain area or is it a consequence of subjective judgement? Help the reader to better understand your interpretation of the model / model component results.

Reply: In our original manuscript, the phrase "differ slightly" was intended to reflect minor variations in PGA values and patterns between the PGA estimates of the two source models. However, we recognize the necessity for a more quantifiable and the range of values will be reported. The text will be modified to acknowledge these differences in a unified manner (% difference).

Regarding the level of knowledge for the reader, it is obvious that the central and the northern parts of Europe have less data, resulting in higher uncertainties for the model. It has been emphasized several times in the manuscript. However, we will ensure that this is further clarified and emphasized in the updated version of the manuscript.

Line 594-600: The European seismic risk model is not based on exactly the same hazard model. Please highlight the major differences otherwise I think this conveys a wrong message.

Reply: A very good observation. The text will be modified accordingly to describe the differences between the two models: i.e., the one used for the calculation of the seismic risk assessment for the Euro-Mediterranean region has a collapsed earthquake rate forecast for each seismogenic source similar to the implementation of ESHM13. Differences are also in the ground motion models used for the risk calculation and the fact that risk computations take into account the modeling of the site response at regional scale, as indicated in Crowley et al (2021), and Weatheril et al (2023)

*Weatherill, G., Crowley, H., Roullé, A., et al.: Modelling site response at regional scale for the 2020 European Seismic Risk Model (ESRM20), Bull Earthquake Eng, 21, 665–714, https://doi.org/10.1007/s10518-022-01526-5, 2023.*

---

## Author Comment (AC2)

Reviewer 2:
Review of "The 2020 European Seismic Hazard Model: Overview and Results" by Danciu et al.

This manuscript gives an overview of an update to the European Seismic Hazard Model. This is a large collaborative effort and the authors succeeded in their goal of building a more comprehensive, updated model. I am especially impressed with their treatment and presentation of epistemic uncertainties throughout the manuscript. I understand that the model is more or less final, and that my comments are most useful as they relate to the presentation of information about the model. The following are comments and requests for clarifications at various points throughout the manuscript. One area that could use improvement is the description of the active crustal fault source model. But, overall, this is a well organized overview paper that understandably can't cover every detail of such a complex community model.

We would like to express our gratitude to the reviewer, who took his time to review and we truly appreciate the constructive feedback on our manuscript. Regarding your comment on the active crustal fault source model, we acknowledge that this section could benefit from additional clarification and detail. It shall be noted that in this special issue, the manuscript of Basili et al 2023 (already cited in other parts) is a companion manuscript depicting details of the active faults database, curation, and harmonization, which we recommend to the readers.

Basili, R., Danciu, L., Beauval, C., Sesetyan, K., Vilanova, S. P., Adamia, S., Arroucau, P., Atanackov, J., Baize, S., Canora, C., Caputo, R., Carafa, M. M. C., Cushing, E. M., Custódio, S., Demircioglu Tumsa, M. B., Duarte, J. C., Ganas, A., García-Mayordomo, J., Gómez de la Peña, L., Gràcia, E., Jamšek Rupnik, P., Jomard, H., Kastelic, V., Maesano, F. E., Martín-Banda, R., Martínez-Loriente, S., Neres, M., Perea, H., Šket Motnikar, B., Tiberti, M. M., Tsereteli, N., Tsironi, V., Vallone, R., Vanneste, K., Zupančič, P., and Giardini, D.: The European Fault-Source Model 2020 (EFSM20): geologic input data for the European Seismic Hazard Model 2020, Nat. Hazards Earth Syst. Sci. Discuss. [preprint], https://doi.org/10.5194/nhess-2023-118, in review, 2023. https://nhess.copernicus.org/preprints/nhess-2023-118/

Lines 77-79: "and it was optimized for large-scale computation of the ground shaking hazard depicted by Peak Ground Acceleration (PGA) and a pseudo-acceleration spectrum (SA) with 5% damping at fifteen spectral ordinates from 0.05s to 5s."

I don't understand what this means, please clarify; how was it optimized?

Reply: By "*optimized*" we refer to improvements made to the ESHM20 computational model, enabling efficient large-scale computation of ground-shaking hazards. This optimization reduces the need for extensive post-processing required in the ESHM13 model. We will correct the text for clarity.

Figure 1: Although this information is available in the caption, it would be nice to include labels for each subpanel so the reader can quickly identify which map is for ESHM13 and ESHM20

Reply: to be corrected accordingly.

Line 161: "methodological enchantments" -- although some may find seismic hazard models enchanting, I think this may be a typo ("enhancements"?).

Reply: to be corrected accordingly.

Lines 208-224: I could be misunderstanding, but I find this section to be odd. It sounds as though you decided to use the Reasenberg (1985) model because it left the most events in (ie., it declustered the least).

Is this because of the papers you mention, such as Marzocchi and Taroni (2014) that argue for not declustering (for purposes of determining earthquake rates; you still need to decluster for the spatial distribution)? That seems like a bad reason to use an antiquated declustering model. At least in the US NSHM23, Reaseberg got the lowest weight for spatial seismicity declustering because it did such a poor job of removing aftershocks (Field et al., 2024; doi: 10.1785/0120230120)--which, again, seems to be the reason that you seem to have chosen it? Because it did the worst job at removing aftershocks and got closest to using no declustering model at all? I know it's too late to change, but I would advocate for using a modern declustering algorithm for the spatial component, and no declustering for the overall rate model. Regardless, you need to more explicitly state why Reasenberg was chosen.

Reply: We understand the confusion and appreciate the opportunity to clarify our approach. Contrary to the misunderstanding, we did not choose the Reasenberg (1985) model for our declustering process. Instead, we utilized the Grünthal (1985) method, which is the default declustering approach for ESHM20. This decision was based on its established application and suitability for the European seismic context, rather than the number of events it retains or excludes.
Your point regarding the application of modern declustering algorithms is well-taken. However, we did not base our choice on the ability of the method to retain more events, but on its historical use, acceptance, and adaptability and calibration of the Grunthal method to the European context.
We will revise this section to more explicitly state why the Grünthal declustering method was chosen and to avoid any confusion regarding the use of the Reasenberg model.

Lines 213-214: "suggested that the cluster method (Reasenberg, 1985) and an alternative to the declustering method (Grünthal, 1985) used in ESHM20" Missing a word here? "Should be used"?

Reply: to be corrected accordingly.

Figure 2: The green volcanic area sources in 2a are barely perceptible. I suggest a larger figure and more distinct color from the gray. Also suggest that you change "(black)" to "(black polygons)" to make it more clear.

Reply: to be corrected accordingly.

Lines 273-283: I find this description to be lacking, it's possible that some of the details are covered in other papers but I believe that they should at least be summarized.

1. Are all faults fully segmented in your model? Multifault ruptures are commonly observed in nature and are included in other leading regional seismic hazard models (UCERF3, NZ NSHM22, US NSHM23). If multifault ruptures are not included in your model, that should be stated and explained. If multisegment ruptures are not included in your model, that is a major issue that affects the usefulness of your model and needs to be explained. Is the 2023 Türkiye-Syria rupture in your model as a single event?

Reply: The faults are not at all segmented and may host earthquake ruptures of any size up to the Mmax at any point along their length and width. The distinction between one fault and another is based on structural geology criteria adopted by a large community of scientists that built and maintained the active fault databases in various regions within Europe and were then blended in EFSM20 (Basili et al., 2023; already cited). This implies that multisegment ruptures are inherently included in the fault model but only within individual fault systems, whereas ruptures jumping from

one fault system to another cannot be modeled because such an option is presently unavailable in OpenQuake. We will add a sentence to clarify this point.

2. You mention that you use slip rates to determine your activity rate parameters; are the slip rates well fit in the final model? In other words, if you take your branch-averaged rate model and compute average slip per rupture and multiply it by the rate of each rupture, then sum those values across all fault patches, does it match the original slip rate?

Reply: Actually, we convert the fault moment rate into activity rate parameters (a-value of the MFDs) using the formulations of Anderson and Luco (1983) and Kagan and Jackson (2000). Therefore, the moment rate is analytically converted into the number of earthquake ruptures per magnitude bin per year and then uniformly distributed over the entire fault area. No fitting procedure is used. We can rephrase the text and/or add a sentence to explain this.

Lines 364-398: I like this description of the computational challenges associated with uncorrelated logic trees, or even large correlated ones. Did you do any sensitivity tests to see how mean hazard computed with your simplified logic tree compares to the full correlated tree?

Reply: Yes, indeed. We conducted sensitivity analyses between the correlated and uncorrelated logic tree branches at the seismogenic source level. Please see the reply to Reviewer A for the same matter. However, further clarification is needed. We did not use a simplified logic tree – we implemented a fully correlated logic tree and used the sampling technique of OpenQuake for the final calculation of the ground-shaking hazard.

Line 409: Missing space in "84thand"

Reply: to be corrected accordingly.

Figure 7: I recommend that you remind the reader somewhere around here of Figure 1 and the electronic supplement in order to see comparison maps with ESHM13. By the timeI got to this figure I had forgotten the earlier mention of the supplement.

Reply: to be corrected accordingly

Figure 9: You need to specify that these are Log10(ratio), not just ratios, both in the caption and above the color scale. I came across this figure before reading the description in the text and thought that I must be going crazy seeing ratios <1.

Reply: to be corrected accordingly

Lines 533-548: I think this section would be cleaner if you stuck to giving linear ratios (even if you plot the log ratios in the figure for more dynamic range), rather than switching back and forth between log and linear (and sometimes omitting the linear).

Reply: to be corrected accordingly

Line 620: Missing space in " 2016)and"

Reply: to be corrected accordingly

Line 625: "These methods are likely to involve the use of physics-based simulations of both earthquake ruptures and/or ground shaking (refs)"

You references examples of physics-based earthquake rupture *or* ground motion simulation, but not an example of both (except for, arguably, Li et al., which is both but for a handful of ruptures and not a probabilistic calculation). I suggest referencing Milner et al. (2021; https://doi.org/10.1785/0120200216), which is the only study I'm aware of that does physics-based earthquake rupture *and* ground motion simulation in a probabilistic calculation.

Thank you for pointing out the omission and for providing the reference to Milner et al. (2021). We appreciate your suggestion and will add this reference to our manuscript to illustrate an example of physics-based earthquake rupture and ground motion simulation used in a probabilistic calculation. Your insight is invaluable to enhancing the accuracy and depth of our paper.

Supplement:

Figures S1 and S2: which model is which in the left panel? Please label. You also need to make the map titles and axis labels larger, they are barely readable (especially for the left panel).

Reply: to be corrected accordingly